# PLCγ-dependent mTOR signalling controls IL-7-mediated early B cell development

Mei Yu[1], Yuhong Chen[1], Hu Zeng [2], Yongwei Zheng[1], Guoping Fu[1], Wen Zhu[1,3], Ulrich Broeckel[4], Praful Aggarwal[4], Amy Turner[4], Geoffrey Neale [5], Cliff Guy[2], Nan Zhu[1], Hongbo Chi[2], Renren Wen[1] & Demin Wang[1,6,7]

The precise molecular mechanism underlying the regulation of early B cell lymphopoiesis is unclear. The PLCγ signaling pathway is critical for antigen receptor-mediated lymphocyte activation, but its function in cytokine signaling is unknown. Here we show that PLCγ1/PLCγ2 double deficiency in mice blocks early B cell development at the pre-pro-B cell stage and renders B cell progenitors unresponsive to IL-7. PLCγ pathway inhibition blocks IL-7-induced activation of mTOR, but not Stat5. The PLCγ pathway activates mTOR through the DAG/PKC signaling branch, independent of the conventional Akt/TSC/Rheb signaling axis. Inhibition of PLCγ/PKC-induced mTOR activation impairs IL-7-mediated B cell development. PLCγ1/PLCγ2 double-deficient B cell progenitors have reduced expression of genes related to B cell lineage, IL-7 signaling, and cell cycle. Thus, IL-7 receptor controls early B lymphopoiesis through activation of mTOR via PLCγ/DAG/PKC signaling, not via Akt/Rheb signaling.

[1] Blood Research Institute, Blood Center of Wisconsin, Milwaukee, WI 53226, USA. [2] Department of Immunology, St. Jude Children's Research Hospital, Memphis, TN 38105, USA. [3] Interdisciplinary Program in Biomedical Sciences, Milwaukee, WI 53226, USA. [4] Section of Genomic Pediatrics, Department of Pediatrics, Medical College of Wisconsin, Milwaukee, WI 53226, USA. [5] Hartwell Center for Bioinformatics and Biotechnology, St. Jude Children's Research Hospital, Memphis, TN 38105, USA. [6] Biomedical Research Center of South China, College of Life Sciences, Fujian Normal University, Fuzhou, Fujian 350117, China. [7] Department of Microbiology and Molecular Genetics, Medical College of Wisconsin, Milwaukee, WI 53226, USA. Correspondence and requests for materials should be addressed to D.W. (email: demin.wang@bcw.edu)

**B** cells are derived from haematopoietic stem cells[1, 2]. Pre-pro-B cells are the earliest B cell progenitors, whereas pro-B cells initiate immunoglobulin heavy (IgH) chain gene rearrangement. A successfully rearranged IgH chain associates with the surrogate L-chains to form the pre-B cell receptor (pre-BCR) that directs the expansion of pre-B cells[1–3]. Deficiency of pre-BCR function blocks early B cell development at the pro-B to pre-B transition[4, 5]. In addition, the interleukin-7 receptor (IL-

7R) has a critical function in early B cell lymphopoiesis[6, 7]. IL-7R signals direct the differentiation of common lymphoid progenitors (CLP) into pro-B cells, support pro-B cell proliferation and survival, regulate IgH gene rearrangement, and promote pre-B cell expansion[6–9]. Deficiency of IL-7 or IL-7R arrests B cell development at the early pre-pro-B stage[6, 7, 10, 11].

IL-7R consists of the specific IL-7 receptor subunit α (IL-7Rα) and the common γ chain (γc)[12–14]. Upon IL-7 binding, the IL-7R

**Fig. 1** PLCγ1/PLCγ2 double deficiency blocks B cell development at the pre-pro-B cell stage. BM from *Plcg1+/+Plcg2+/+*, *Plcg1−/−Plcg2+/+*, *Plcg1+/+Plcg2−/−*, or *Plcg1−/−Plcg2−/−* mice were transplanted into lethally irradiated congenic wild-type CD45.1+ mice. Six to eight weeks after transplantation, the recipients were analyzed. **a** BM cells from the recipients are largely YFP+ donor-derived. Lin−CD45.2+ BM cells from the recipients were analyzed. Numbers indicate percentages of YFP+ cells in the gated live cells. **b** Both PLCγ1 and PLCγ2 are deleted in YFP+ BM cells from *Plcg1−/−Plcg2−/−* mice. YFP+ BM cells from the recipients were subjected to western blot analysis. **c** Both PLCγ1 and PLCγ2 are expressed in pro-B cells. Sorted pro- (B220loIgM−CD43+) and FO (B220+IgMloCD23+CD21int) B cells from wild-type mice were subjected to western blot analysis. **d** BM cells from the recipients were stained with anti-CD45.2, anti-B220, and anti-IgM. Numbers indicate percentages of pro/pre-, immature, and mature B cells in the gated CD45.2+ population. **e** Bar graphs show the numbers of CD45.2+ pro/pre-, immature, and mature B cells in BM. **f** BM cells from the recipients were stained with anti-CD45.2, anti-B220, anti-IgM, anti-CD25, and anti-CD43. Numbers indicate percentages of pro-B (upper) and pre-B (lower) cells in the gated CD45.2+B220+IgM− population. **g** Bar graphs show the numbers of CD45.2+ pro-B cells and pre-B cells in BM. **h** BM cells from the recipients were stained with anti-CD45.2, anti-B220, anti-CD43, anti-CD24, and anti-BP-1. BM from µMT and Jak3-deficient mice served as the controls. Numbers indicate percentages of pre-pro-B (fraction A), early pro-B (fraction B) and late pro-B/early pre-B (fraction C/C′) cells in the gated CD45.2+B220+CD43+ pro-B cell population. **i** Bar graphs show the numbers of CD45.2+ fractions A, B, and C/C′ B cells in BM. Error bars show ± SEM. Data shown are obtained from or representative of 10 (**a**, **d**–**g**) or 8 (**h**, **i**) mice of each genotype or representative of 2 (**b**, **c**) independent experiments. Control data are representative of 3 µMT and Jak3-deficient mice

heterodimer activates Jak1 and Jak3, leading to the activation of Stat5 proteins[15, 16]. Stat5 in turn translocates to the nucleus and activates transcription of a variety of genes[17–19]. Disruption of IL-7Rα, γc, Jak1 or Jak3, or Stat5 impairs IL-7-mediated early lymphoid development, resulting in arrest of B cell development at the pre-pro-B cell stage[6, 20–25], highlighting central functions for the Jak1/Jak3/Stat5 pathway in IL-7-mediated early B cell development. Despite considerable progress, IL-7R signaling pathways that control early B cell development are not fully understood.

Phospholipase Cγ (PLCγ) is a lipid-hydrolyzing enzyme that upon activation hydrolyzes phosphatidyl-inositol 4,5-bisphosphate ($PIP_2$) to generate diacylglycerol (DAG) and inositol 1,4,5-trisphosphate ($IP_3$)[26, 27]. DAG activates protein kinase C (PKC) to turn on the protein kinase TAK1 through the CARMA1/Bcl10/MALT1 ternary complex, leading to the activation of IκB kinase (IKK) and c-Jun N-terminal protein kinase (JNK)[28–31]. IKK activates the transcription factor NF-κB[32, 33], whereas JNK results in the activation of the transcription factor AP1[34, 35]. $IP_3$ induces increase of the intracellular $Ca^{2+}$ concentration, leading to the activation of the phosphatase calcineurin. Calcineurin dephosphorylates multiple phosphoserines on NFAT, leading to its nuclear translocation and activation[36].

PLCγ has two isoforms, γ1 and γ2[26, 27]. PLCγ2 is required for many aspects of BCR-mediated signaling and late B cell maturation and function[31, 37–41]. PLCγ1 is essential for T cell receptor (TCR)-mediated signaling and T cell development[42, 43]. Although the individual functions of PLCγ1 and PLCγ2 have been identified, potential overlapping functions of the two PLCγ isoforms could mask the central biological function of the PLCγ pathway.

Here, we generate PLCγ1/PLCγ2 double-deficient mice and show that both PLCγ isoforms have critical and redundant functions in early lymphopoiesis. PLCγ1/PLCγ2 double deficiency blocks early B cell development at the pre-pro-B cell stage and T cell development at the double negative (DN) stage. Importantly, PLCγ1/PLCγ2 double-deficient B cell progenitors do not respond to IL-7, demonstrating a critical function of the PLCγ pathway in response to IL-7 in vivo. Moreover, the PLCγ pathway has no effect on Stat5 activation, but directly controls AKT-independent activation of mTOR. Thus, our studies, for the first time, show that the PLCγ pathway is essential for cytokine receptor-mediated biological function and that IL-7R activates mTOR through an unconventional PLCγ/DAG/PKC-dependent pathway to control early B lymphopoiesis.

## Results

**PLCγ1/PLCγ2 double deficiency blocks early B cell development**. To overcome embryonic lethality caused by PLCγ1 deficiency and thus to study the combined role of both PLCγ1 and PLCγ2 in early B cell development, we crossed *Plcg1* "floxed" mice (*Plcg1*$^{fl/+}$) with heterogeneous PLCγ2-deficient (*Plcg2*$^{+/-}$) mice and *MxCre* transgenic mice, in which Cre expression is under the control of type I IFN-inducible Mx promoter. To track deletion of the "floxed" *Plcg1* gene, the mice were further bred with *Rosa-YFP* transgenic mice, in which a YFP cDNA preceded by a floxed transcriptional stop cassette is inserted into the ubiquitously expressed rosa26 locus[44], to generate *YFPMxCrePlcg1*$^{fl/fl}$*Plcg2*$^{-/-}$ mice. To study haematopoietic cell intrinsic effect of PLCγ1/PLCγ2 double deficiency, bone marrow (BM) cells from poly(I-C)-treated *YFPMxCrePlcg1*$^{+/+}$*Plcg2*$^{+/+}$ (*Plcg1*$^{+/+}$*Plcg2*$^{+/+}$), *YFPMxCrePlcg1*$^{fl/fl}$*Plcg2*$^{+/+}$ (*Plcg1*$^{-/-}$*Plcg2*$^{+/+}$) or *YFPMxCrePlcg1*$^{+/+}$*Plcg2*$^{-/-}$ (*Plcg1*$^{+/+}$*Plcg2*$^{-/-}$) control mice, or *YFPMxCrePlcg1*$^{fl/fl}$*Plcg2*$^{-/-}$ (*Plcg1*$^{-/-}$*Plcg2*$^{-/-}$) experimental mice were transplanted into lethally irradiated congenic wild-type

$CD45.1^+$ recipients. Six to eight weeks after BM transplantation, BM cells were largely $CD45.2^+YFP^+$ in the recipients that received the BM of control or experimental mice (Fig. 1a). $YFP^+$ BM cells from the recipients of BM from *Plcg1*$^{-/-}$*Plcg2*$^{-/-}$ mice had no expression of PLCγ1 or PLCγ2, whereas $YFP^+$ BM cells from the control recipients expressed both or either one of the two PLCγs (Fig. 1b). YFP positivity in BM cells from the recipients of BM of *Plcg1*$^{-/-}$*Plcg2*$^{+/+}$ or *Plcg1*$^{-/-}$*Plcg2*$^{-/-}$ mice was tightly correlated with PLCγ1 deletion (Fig. 1b). Of note, both PLCγ1 and PLCγ2 were completely deleted in $CD45.2^+YFP^+$ BM cells in the recipients of BM from *Plcg1*$^{-/-}$*Plcg2*$^{-/-}$ mice (Fig. 1b); however, the numbers of the $CD45.2^+$ hematopoietic stem cell (HSC)-containing Lin$^-$c-Kit$^+$Sca1$^+$ (LSK) or myeloid progenitor-containing Lin$^-$c-Kit$^+$Sca1$^-$ (LK) cells were normal in the recipients of BM from *Plcg1*$^{-/-}$*Plcg2*$^{-/-}$ relative to *Plcg1*$^{+/+}$*Plcg2*$^{+/+}$, *Plcg1*$^{-/-}$*Plcg2*$^{+/+}$, or *Plcg1*$^{+/+}$*Plcg2*$^{-/-}$ mice (Supplementary Fig. 1a). Owing to the reduction of B cells, the percentages of $CD45.2^+$Mac-1$^+$Gr-1$^+$ myeloid cells were increased in the recipients of BM from *Plcg1*$^{-/-}$*Plcg2*$^{-/-}$ relative to the control mice (Supplementary Fig. 1b). However, the numbers of $CD45.2^+$Mac-1$^+$Gr-1$^+$ myeloid cells were normal in the recipients of BM from *Plcg1*$^{-/-}$*Plcg2*$^{-/-}$ mice (Supplementary Fig. 1c). The expression of CD11c, F4/80, and c-fms was normal on $YFP^+CD45.2^+$ myeloid cells in the recipients of BM from *PLCγ1*$^{-/-}$*PLCγ2*$^{-/-}$ relative to the control mice (Supplementary Fig. 1d, e). In addition, in vitro cytokine-driven colony formation assay revealed that *Plcg1*$^{-/-}$*Plcg2*$^{-/-}$ BM cells responded normally to SCF and M-CSF to form colonies (Supplementary Fig. 1f). After 9 days of co-culture with OP9 stromal cells in the presence of SCF, *Plcg1*$^{-/-}$*Plcg2*$^{-/-}$ Lin$^-$ BM progenitors generated similar amount of $CD45.2^+$Mac-1$^+$ myeloid cells as *Plcg1*$^{+/+}$*Plcg2*$^{+/+}$, *Plcg1*$^{-/-}$*Plcg2*$^{+/+}$, *Plcg1*$^{+/+}$*Plcg2*$^{-/-}$, *Rag1*$^{-/-}$, or *Jak3*$^{-/-}$ progenitors (Supplementary Fig. 1g). Taken together, both PLCγ1 and PLCγ2 can be completely deleted in the hematopoietic compartment and PLCγ1/PLCγ2 double deficiency has no effect or minor on the populations of HSCs and myeloid lineage cells.

The expression levels of both PLCγ1 and PLCγ2 were high in pro-B cells, whereas PLCγ2 was the dominant isoform in peripheral follicular B cells (Fig. 1c), indicating a combined role of both PLCγs in early B cell development. We examined the effect of PLCγ1/PLCγ2 double deficiency on B cell development. Within the $YFP^+$ BM cells, the populations of B220$^+$ B cells as well as immature (B220$^+$IgM$^+$) and mature (B220$^{hi}$IgM$^+$) B cells were barely detectable in the recipients of BM from *Plcg1*$^{-/-}$*Plcg2*$^{-/-}$ mice (Fig. 1d, e). The remaining residual B220$^+$ cells were mainly B220$^+$IgM$^-$ pro/pre-B progenitors that were markedly reduced in the recipients of BM from *Plcg1*$^{-/-}$*Plcg2*$^{-/-}$ mice (Fig. 1d, e). In contrast, the populations of pro/pre-, immature and mature in the recipients of *Plcg1*$^{+/+}$*Plcg2*$^{+/+}$ or *Plcg1*$^{-/-}$*Plcg2*$^{+/+}$ BM were normal (Fig. 1d, e). As expected[37], the populations of pro/pre- and immature B cells were normal whereas that of mature B cells was markedly reduced in the recipients of *Plcg1*$^{+/+}$*Plcg2*$^{-/-}$ BM (Fig. 1d, e). CD43 is expressed on B cell progenitors prior to pre-B cells, whereas CD25 is only expressed on pre-B cells[2, 45]. Within the B220$^+$IgM$^-$ progenitors, the recipients of *Plcg1*$^{-/-}$*Plcg2*$^{-/-}$ BM exhibited lower but significant amount of B220$^+$CD43$^+$ B cell progenitors but hardly detectable B220$^+$CD25$^+$ pre-B cells, compared to the recipients of *Plcg1*$^{+/+}$*Plcg2*$^{+/+}$, *Plcg1*$^{-/-}$*Plcg2*$^{+/+}$, or *Plcg1*$^{+/+}$*Plcg2*$^{-/-}$ BM (Fig. 1f, g). Of note, the control recipients of *Plcg1*$^{+/+}$*Plcg2*$^{-/-}$ BM had a reduction of B220$^+$CD25$^+$ pre-B cells (Fig. 1f, g). Further, B220$^+$CD43$^+$ B cell progenitors can be divided into B220$^+$CD43$^+$BP-1$^-$CD24$^-$ pre-pro-B (fraction A), B220$^+$CD43$^+$BP-1$^-$CD24$^+$ early pro-B (fraction B), and B220$^+$CD43$^+$BP-1$^+$CD24$^{+/hi}$ late pro-B and early pre-B (fraction C/C′) cells[2, 46]. Within the B220$^+$CD43$^+$ cells, the recipients of *Plcg1*$^{-/-}$*Plcg2*$^{-/-}$ relative to

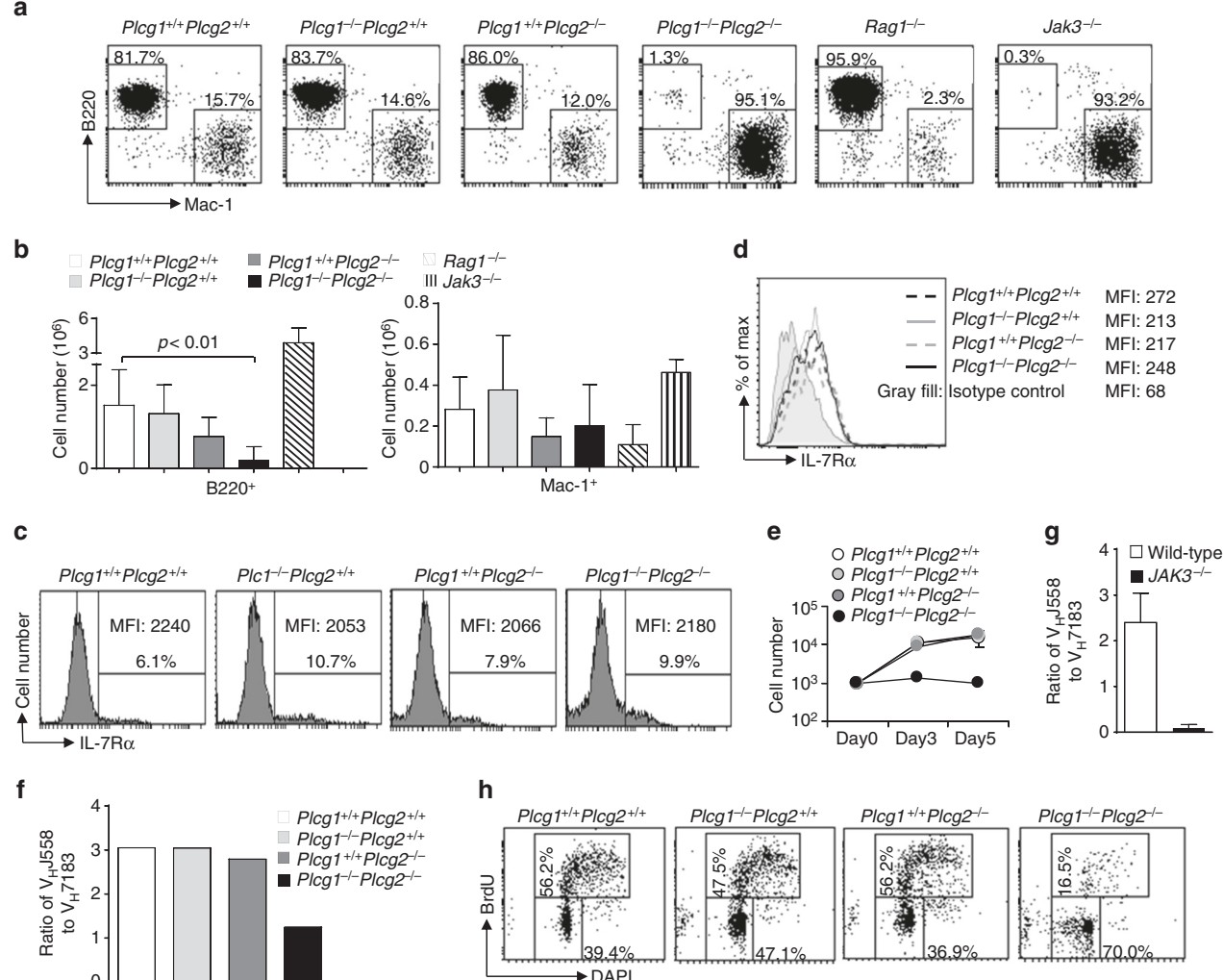

**Fig. 2** PLCγ1/PLCγ2 double deficiency impairs IL-7-mediated B cell progenitor functions. BM from $Plcg1^{+/+}Plcg2^{+/+}$, $Plcg1^{-/-}Plcg2^{+/+}$, $Plcg1^{+/+}Plcg2^{-/-}$, or $Plcg1^{-/-}Plcg2^{-/-}$ mice were transplanted into lethally irradiated congenic wild-type CD45.1[+] mice. Six to eight weeks after transplantation, Lin[−] BM cells from the recipients, $Rag1^{-/-}$ or $Jak3^{-/-}$ mice were cultured on OP9 cells plus IL-7. **a, b** IL-7-dependent B cell production. The cells were stained with anti-CD45.2, anti-B220, and anti-Mac-1 after 9 days of co-culture. Numbers indicate percentages of B220[+] and Mac-1[+] cells in the gated CD45.2[+] population (**a**). Bar graphs show the numbers of CD45.2[+]B220[+] and CD45.2[+]Mac-1[+] cells (**b**). **c, d** IL-7R expression. BM cells from the recipients were stained with anti-IL-7Rα. Numbers indicate percentages of IL-7Rα[+] cells in the gated CD45.2[+]Lin[−]population (**c**). After 9 days of co-culture plus Flt-3L, the cells from the recipients were stained with anti-CD45.2, anti-B220, and anti-IL-7Rα. Levels of IL-7Rα expression in the gated CD45.2[+]B220[+] population are shown (**d**). **e** Expansion in response to IL-7. Fractions B and C of B cell progenitors (YFP[+]CD45.2[+]B220[+]CD43[+]CD24[+]) were sorted from the recipients and cultured with OP9 plus IL-7. On the indicated days after co-culture, the numbers of live cells were determined by trypan blue exclusion. **f, g** IL-7-mediated rearrangement of the distal $V_H$J558 family gene segments of IgH chain genes. YFP[+]CD45.2[+]B220[+]CD43[+] or B220[+]IgM[−]CD43[+] pro-B cells were sorted from the recipients (**f**) or from wild-type or Jak3[−/−] mice (**g**), respectively. The frequency of $V_H$J558 and $V_H$7183 family usage in these progenitors was determined by high-throughput sequencing of the rearranged IgH genes. Bar graphs show the ratios of percentages of rearranged $V_H$J558 genes to those of rearranged $V_H$7183 genes in the pro-B cells. **h** IL-7-mediated B cell progenitor proliferation. YFP[+]CD45.2[+]B220[+]CD43[+]CD24[+] B cell progenitors (fractions B and C) were sorted from the recipients, and co-cultured with OP9 cells plus IL-7 for 3 days. BrdU incorporation and DAPI staining were used to determine proliferation. Numbers indicate percentages of BrdU[+] cells in the gated live cells. Error bars show ± SEM. Data shown are obtained from or representative of 6 (**a, b**), 4 (**c, h**), or 2 (**d–f**) independent experiments or from three wild-type and 2 $Jak3^{-/-}$ mice (**g**)

$Plcg1^{+/+}Plcg2^{+/+}$ or $Plcg1^{-/-}Plcg2^{+/+}$ BM displayed drastic reduction in early pro-B (fraction B) and late pro-B/early pre-B (fraction C/C′) cells (Fig. 1h, i). Consistent with the previous finding[40], the recipients of $Plcg1^{+/+}Plcg2^{-/-}$ BM exhibited an increase in late pro-B/early pre-B (fraction C/C′) cells (Fig. 1h, i). Of note, we also examined B cell development in YFPMx-CrePlc1[fl/fl]Plcg2[−/−] ($Plcg1^{-/-}Plcg2^{-/-}$) and control mice three weeks after the last poly(I-C) injection. Early B cell development was blocked at the pre-pro-B stage in Cre-activated non-transplanted $Plcg1^{-/-}Plcg2^{-/-}$ mice (Supplementary Fig. 2).

Thus, unlike PLCγ2 single deficiency, PLCγ1/PLCγ2 double deficiency blocks B cell development at the pre-pro-B stage.

Signals from both the IL-7R and pre-BCR are critical for early B cell lymphopoiesis at the pro/pre-B cell stages[3, 47]. Disruption of IL-7R or pre-BCR function arrests B cell development at the early pre-pro-B (fraction A) or late pro- and pre-B cell (fraction C/C′) stage, respectively[5, 6]. The recipients of $Plcg1^{-/-}Plcg2^{-/-}$ BM displayed B cell development arrest at the pre-pro-B (fraction A) stage, similar to $Jak3$-deficient mice that have defective IL-7R signaling[22, 23] but not μMT mice that lack pre-BCR signaling[4]

(Fig. 1h). These data indicate that both PLCγ1 and PLCγ2 may be required for IL-7R-mediated early B cell development.

Signals from IL-7R also direct early T cell development at the DN stage. We examined early T cell development in the experimental and control recipients. The number of total thymocytes was markedly reduced in the recipients of BM from $Plcg1^{-/-}Plcg2^{+/+}$ relative to $Plcg1^{+/+}Plcg2^{+/+}$ or $Plcg1^{+/+}Plcg2^{-/-}$ mice, and was further reduced and became barely detectable in the recipients of BM from $Plcg1^{-/-}Plcg2^{-/-}$ relative to $Plcg1^{-/-}Plcg2^{+/+}$ mice (Supplementary Fig. 3a). The number of $CD45.2^+$ DN thymocytes was reduced in the recipients of BM from $Plcg1^{-/-}Plcg2^{+/+}$ relative to $Plcg1^{+/+}Plcg2^{+/+}$ or $Plcg1^{+/+}Plcg2^{-/-}$ mice, but barely detectable in the recipients of BM from $Plcg1^{-/-}Plcg2^{-/-}$ mice (Supplementary Fig. 3b). Further, the number of $CD45.2^+$ DN4 was reduced in the recipients of BM from $Plcg1^{-/-}Plcg2^{+/+}$ relative to $Plcg1^{+/+}Plcg2^{+/+}$ or $Plcg1^{+/+}Plcg2^{-/-}$ mice, whereas the numbers of $CD45.2^+$ DN1, DN2, DN3, and DN4 were all further reduced in the recipients of BM from $Plcg1^{-/-}Plcg2^{-/-}$ relative to $Plcg1^{-/-}Plcg2^{+/+}$ mice (Supplementary Fig. 3c). Moreover, the number of CLPs and BLPs (B cell-biased lymphoid progenitor) was significantly reduced in the recipients of BM from $Plcg1^{-/-}Plcg2^{-/-}$, but not $Plcg1^{-/-}Plcg2^{+/+}$ or $Plcg1^{+/+}Plcg2^{-/-}$, relative to $Plcg1^{+/+}Plcg2^{+/+}$ mice (Supplementary Fig. 4). Of note, deficiency of IL-7 or IL-7R results in marked reduction in the numbers of total DN, DN1, DN2, DN3, and DN4 cells[6, 7, 48, 49]. Deficiency of IL-7 also leads to reduction in the numbers of CLPs[50]. The similarity between the effect of PLCγ1/PLCγ2 double deficiency and that of IL-7 or IL-7R deficiency on early T cell development and CLPs is also consistent with the notion that both PLCγ1 and PLCγ2 may be required for IL-7R-mediated function.

**PLCγ1/γ2 double deficiency impairs IL-7-mediated functions.** To determine whether PLCγ1 and PLCγ2 are directly involved in IL-7-mediated early B cell development, we employed the in vitro OP9 co-culture system. Lin$^-$ BM progenitors from poly(I-C)-treated YFPMxCre $Plcg1^{+/+}Plcg2^{+/+}$, $Plcg1^{-/-}Plcg2^{+/+}$, $Plcg1^{+/+}Plcg2^{-/-}$, or $Plcg1^{-/-}Plcg2^{-/-}$ mice were cultured on OP9 stromal cells with IL-7. After 9 days of culture, Lin$^-$ BM progenitors from $Plcg1^{+/+}Plcg2^{+/+}$, $Plcg1^{-/-}Plcg2^{+/+}$, and $Plcg1^{+/+}Plcg2^{-/-}$ mice developed into mainly B cells and some myeloid cells (Fig. 2a, b). In contrast, Lin$^-$ BM progenitors from $Plcg1^{-/-}Plcg2^{-/-}$ mice failed to give rise to B cells but produced $Mac1^+$ myeloid cells (Fig. 2a, b). As expected, Lin$^-$ BM progenitors from Jak3-deficient mice could develop into myeloid cells but not B cells, whereas those progenitors from Rag1-deficient mice developed into mainly B cells and some myeloid cells (Fig. 2a, b). These data demonstrate that production of B cells from BM progenitors on OP9 cells is IL-7R- but not pre-BCR-dependent, and PLCγ1/PLCγ2 double deficiency inhibits IL-7R-dependent B cell but not IL-7R-independent myeloid cell production.

Further, we examined the effect of acute in vitro deletion of $PLCgs$ on IL-7-mediated B cell production. Lin$^-$ BM progenitors from ERCre, a tamoxifen-inducible form of Cre, $Plcg1^{+/+}Plcg2^{+/+}$, $Plcg1^{fl/fl}Plcg2^{+/+}$, $Plcg1^{+/+}Plcg2^{-/-}$, or $Plcg1^{fl/fl}Plcg2^{-/-}$ mice were cultured on OP9 stromal cells with IL-7 and Flt-3L. After 5 days of culture, Lin$^-$ BM progenitors from all genotype mice developed into B cells and myeloid cells (Supplementary Fig. 5a). These cells were further cultured on OP9 in the presence of IL-7 without or with 4-hydroxytamoxifen (4-OHT) for 4 more days. Without 4-OHT treatment, $Plcg1^{+/+}Plcg2^{+/+}$, $Plcg1^{fl/fl}Plcg2^{+/+}$, $Plcg1^{+/+}Plcg2^{-/-}$ and to a lesser extent $Plcg1^{fl/fl}Plcg2^{-/-}$ cells further developed into B cells (Supplementary Fig. 5b). Following 4-OHT treatment, $Plcg1^{+/+}Plcg2^{+/+}$, $Plcg1^{fl/fl}Plcg2^{+/+}$, and $Plcg1^{+/+}Plcg2^{-/-}$ cells developed into largely B cells and a few myeloid

cells, whereas $Plcg1^{fl/fl}Plcg2^{-/-}$ cells developed into myeloid cells but failed to give rise to B cells (Supplementary Fig. 5c). These data demonstrate that PLCγ1/PLCγ2 double deficiency resulting from acute deletion of the $Plcgs$ genes in vitro blocks IL-7R-dependent B cell but not IL-7R-independent myeloid cell production.

We next examined whether PLCγ1/PLCγ2 double deficiency affected the expression of IL-7 receptor (IL-7R). Despite the reduction of the number of IL-7Rα$^+$ cells, the remaining IL-7Rα$^+$ cells from $Plcg1^{-/-}Plcg2^{-/-}$ BM expressed the same level of IL-7Rα as IL-7Rα$^+$ BM cells from $Plcg1^{+/+}Plcg2^{+/+}$, $Plcg1^{-/-}Plcg2^{+/+}$, and $Plcg1^{+/+}Plcg2^{-/-}$ mice did (Fig. 2c). In addition, the level of IL-7Rα expression on $Plcg1^{-/-}Plcg2^{-/-}$ B cell progenitors derived from co-culture of BM with OP9 cells was similar to that on the corresponding $Plcg1^{+/+}Plcg2^{+/+}$, $Plcg1^{-/-}Plcg2^{+/+}$, and $Plcg1^{+/+}Plcg2^{-/-}$ B cell progenitors (Fig. 2d). Thus, the failure of $Plcg1^{-/-}Plcg2^{-/-}$ BM progenitors to differentiate into B cells in response to IL-7 in vitro was not due to the lack of IL-7R expression.

Further, the residual $B220^+CD43^+CD24^+$ B cell progenitors that contained early pro-B (fraction B) and late pro-B (fraction C) cells were sorted from $Plcg1^{-/-}Plcg2^{-/-}$ mice. These mutant B cell progenitors could not expand following 3 to 5 days of co-culture on OP9 cells with IL-7, whereas those B cell progenitors sorted from $Plcg1^{+/+}Plcg2^{+/+}$, $Plcg1^{-/-}Plcg2^{+/+}$, or $Plcg1^{+/+}Plcg2^{-/-}$ mice expanded well under the same condition (Fig. 2e). Thus, PLCγ1/PLCγ2 double-deficient B cell progenitors are unable to respond to IL-7 in vitro, further indicating a critical role of the PLCγ pathway in IL-7-mediated expansion of B cell progenitors.

One important in vivo function of IL-7R signaling is to direct the rearrangement of the $V_H$J558 family of immunoglobulin (Ig) heavy (H) chain variable ($V_H$) gene segments, which locate furthest away from the diversity ($D_H$) and joining ($J_H$) gene segments at the 5′ end of the $V_H$ cluster of the IgH locus[9]. IL-7R deficiency leads to markedly decreased frequency of $V_H$J558 family usage in rearranged IgH chain genes[9]. To further examine whether PLCγ1 and PLCγ2 have a role in IL-7R-mediated function in vivo, we studied the effect of PLCγ1/PLCγ2 double deficiency on $V_H$J558 family usage. $B220^+CD43^+$ pro-B cells were sorted from $Plcg1^{-/-}Plcg2^{-/-}$ mice and the frequency of $V_H$J558 family usage in these progenitors was determined by high-throughput sequencing of the rearranged IgH genes. The usage frequency of $V_H$J558 family genes relative to proximal $V_H$7183 genes was markedly reduced in pro-B cells from $Plcg1^{-/-}Plcg2^{-/-}$ relative to $Plcg1^{+/+}Plcg2^{+/+}$, $Plcg1^{-/-}Plcg2^{+/+}$ or $Plcg1^{+/+}Plcg2^{-/-}$ mice (Fig. 2f). This defect was similar to that in Jak3-deficient pro-B cells (Fig. 2g). Thus, PLCγ1/PLCγ2 double deficiency indeed impairs IL-7R signaling in vivo, resulting in a reduction of the rearrangement of the distal $V_H$J558 family gene segments.

**The PLCγ pathway supports IL-7-mediated cell proliferation.** To understand the mechanism by which the PLCγ pathway regulates the function of IL-7, we studied the roles of PLCγ1 and PLCγ2 in IL-7-mediated proliferation and survival of B cell progenitors. Residual $YFP^+CD45.2^+B220^+CD43^+CD24^+$ B cell progenitors (early pro-B and late pro-B) were sorted from $Plcg1^{-/-}Plcg2^{-/-}$ mice, and co-cultured with OP9 cells in the presence of IL-7. BrdU incorporation assay showed that the $Plcg1^{-/-}Plcg2^{-/-}$ B progenitors exhibited a decrease in the level of IL-7-induced BrdU uptake compared to the corresponding $Plcg1^{+/+}Plcg2^{+/+}$, $Plcg1^{-/-}Plcg2^{+/+}$, $Plcg1^{+/+}Plcg2^{-/-}$ cells (Fig. 2h). In contrast, TUNEL analysis did not reveal any significant differences in the rate of apoptosis between the $Plcg1^{-/-}Plcg2^{-/-}$ and $Plcg1^{+/+}Plcg2^{+/+}$ B progenitors (Supplementary Fig. 6a). Of note, $Plcg1^{+/+}Plcg2^{+/+}$, $Plcg1^{-/-}Plcg2^{+/+}$, $Plcg1^{+/+}Plcg2^{-/-}$, but not $Plcg1^{-/-}Plcg2^{-/-}$, BM cells produced B cell progenitors when cultured with IL-

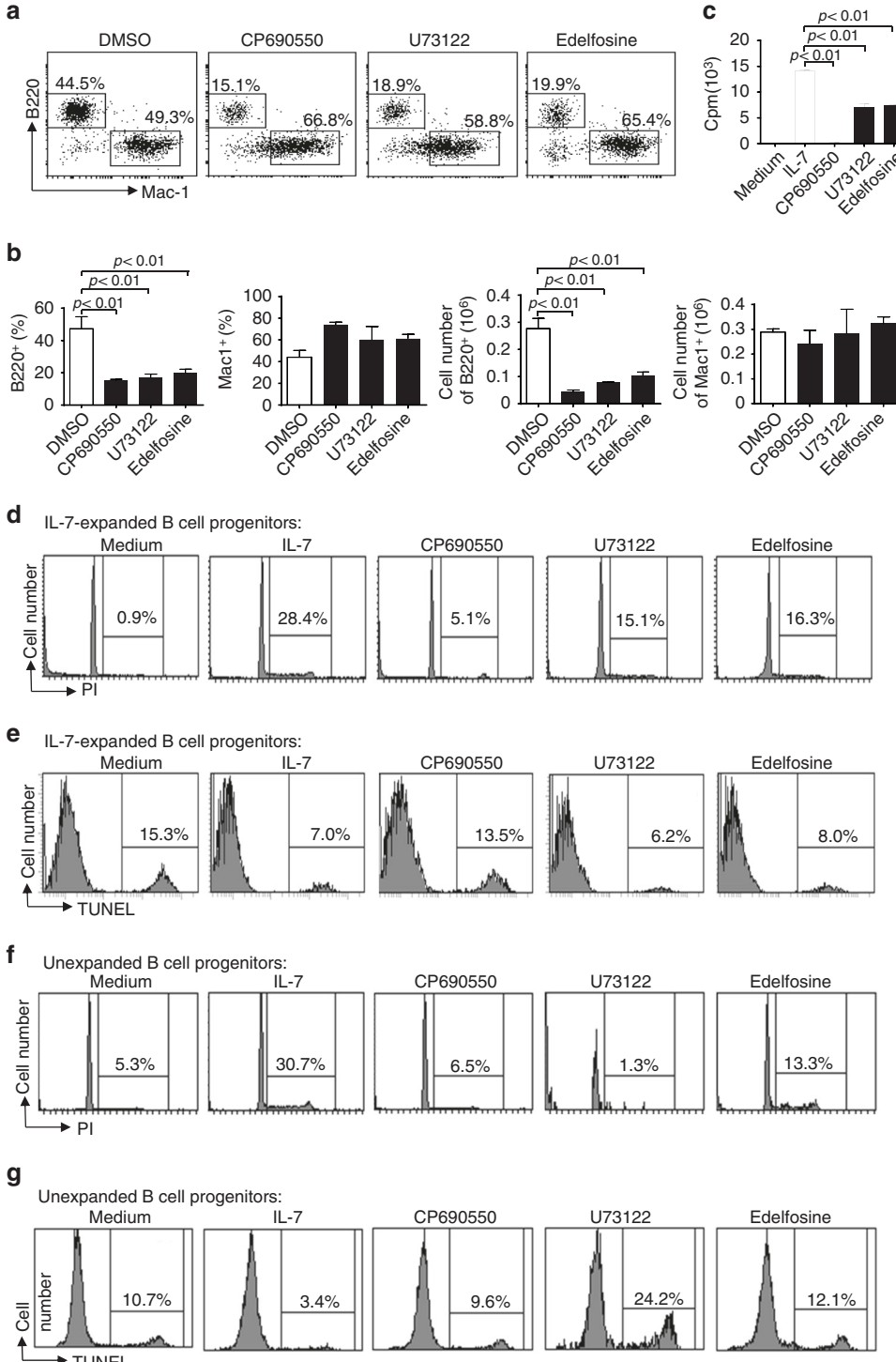

**Fig. 3** Effect of inhibition of the PLCγ pathway on IL-7-dependent B cell functions. **a, b** PLCγ pathway inhibition impairs IL-7-dependent B cell production. Lin⁻ BM cells from wild-type mice were cultured on OP9 cells in the presence of IL-7 for 1 day and then the cells were treated with DMSO, CP690550, U73122, or edelfosine. The cells were stained with anti-B220 and anti-Mac-1. Numbers indicate percentages of B220$^+$ and Mac-1$^+$ cells in the gated live cells (**a**) and bar graphs show the percentages and numbers of B220$^+$ and Mac-1$^+$ cells (**b**). **c-e** PLCγ pathway inhibition impairs IL-7-mediated proliferation but not survival of in vitro-expanded B cell progenitors. BM cells from *Rag1*-deficient mice were cultured in the presence of IL-7 for 7 days to derive B cell progenitors. These B cell progenitors were then cultured without IL-7 (medium) or with IL-7 plus DMSO, CP690550, U73122, or edelfosine. Cell proliferative responses were determined by [³H] thymidine incorporation (**c**), cell-cycle analysis was performed by PI staining (**d**), and cell survival was measured by TUNEL staining (**e**). **f, g** PLCγ pathway inhibition impairs IL-7-mediated proliferation and survival of unexpanded B cell progenitors. B220$^+$IgM$^-$IL-7R$^+$ B cell progenitors were sorted from wild-type mice and cultured without IL-7 (medium) or with IL-7 plus DMSO, CP690550, U73122, or edelfosine. Cell-cycle analysis was performed by PI staining (**f**) and cell survival was measured by TUNEL staining (**g**). Error bars show ± SEM. Data shown are obtained from or representative of 3 (**a, b**), 6 (**c**), 7 (**d, e**), and 5 (**f, g**) independent experiments

7 alone (Supplementary Fig. 6b). However, all $Plcg1^{+/+}Plcg2^{+/+}$, $Plcg1^{-/-}Plcg2^{+/+}$, $Plcg1^{+/+}Plcg2^{-/-}$, and $Plcg1^{-/-}Plcg2^{-/-}$ BM cells failed to generate B cells when cultured with OP9 stromal cells alone. Thus, PLCγ1/PLCγ2 double deficiency impairs IL-7-mediated proliferation of B cell progenitors.

To confirm that the PLCγ pathway is critical for IL-7-mediated proliferation, we examined the effect of PLCγ inhibitor on IL-7-induced B cell production from Lin⁻ BM progenitors in vitro. In the presence of U73122, an aminosteroid PLCγ inhibitor, or edelfosine, a lysophospholipid PLCγ inhibitor, wild-type BM

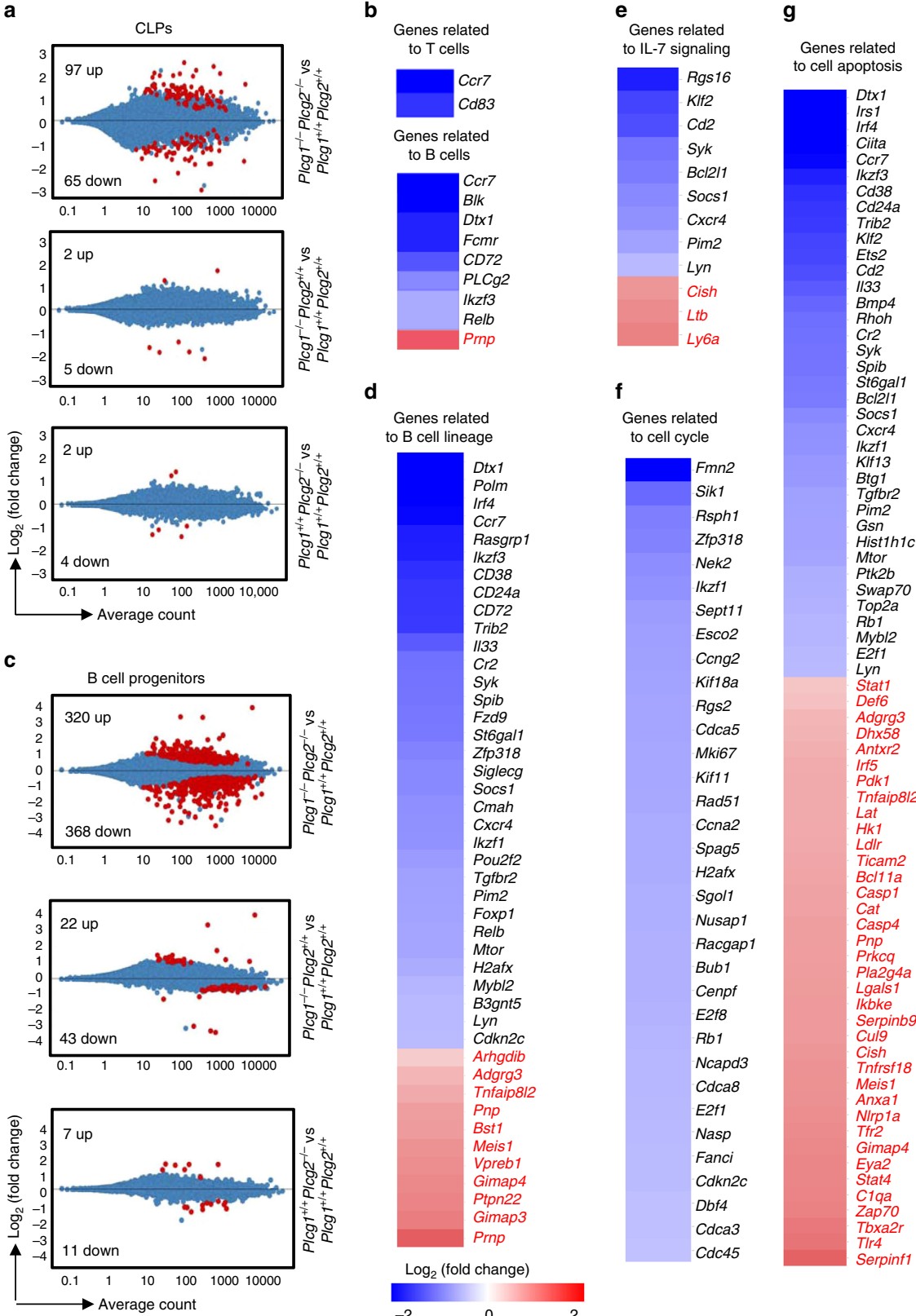

progenitors mainly developed into Mac1$^+$ myeloid instead of B cells following IL-7-mediated co-culture with OP9 cells (Fig. 3a, b). As expected, the Jak3 inhibitor CP-690550 (tofacitinib) inhibited wild-type BM progenitors from developing into B but not myeloid cells in response to IL-7 on OP9 cells (Fig. 3a, b). In contrast, in the absence of any inhibitors, wild-type BM progenitors largely developed into B cells with some myeloid cells in response to IL-7 (Fig. 3a, b). These B cells were mostly B220$^+$CD43$^+$BP-1$^-$CD24$^+$ early pro-B (Supplementary Fig. 6c). CP-690550 or U73122 effectively reduced the expansion of these B cells (Supplementary Fig. 6c). Thus, inhibition of PLCγ, similar to that of Jak3, blocks IL-7-dependent B cell but not IL-7-independent myeloid cell production.

Consistently, U73122, edelfosine, or CP-690550 markedly reduced the rate of IL-7-induced $^3$H-thymidine incorporation of B cell progenitors derived from BM as compared to the control cells in the absence of the inhibitors (Fig. 3c). Further, propidium iodide (PI) staining in combination with FACS analysis revealed that U73122, edelfosine, or CP-690550 impaired IL-7-induced entry of these B cell progenitors into S and G2/M phases (Fig. 3d). The TUNEL assay revealed that IL-7 decreased the population of apoptotic cells in B cell progenitors as compared to the control cells with medium alone (Fig. 3e). Of note, CP-690550 but not U73122 or edelfosine increased the population of apoptotic cells in B cell progenitors in the presence of IL-7 as compared with the IL-7-cultured control cells without any inhibitors (Fig. 3e). In addition, manoalide, a potent PLC inhibitor, impaired IL-7-induced entry of the B cell progenitors into S and G2/M phases (Supplementary Fig. 6d) and increased the population of apoptotic cells in IL-7-cultured B cell progenitors (Supplementary Fig. 6e) as compared to the control cells without any inhibitors. These data demonstrate that inhibition of the PLCγ pathway impairs IL-7-mediated proliferation and possibly reduces survival of in vitro-expanded B cell progenitors.

Furthermore, we examined the effect of PLCγ inhibitor on IL-7-mediated proliferation and survival of unexpanded B cell progenitors. IL-7R$^+$ B cell progenitors were sorted from wild-type mice and then directly cultured without IL-7 or with IL-7 plus DMSO or the inhibitors. U73122, edelfosine, or CP-690550 impaired IL-7-induced entry of these B cell progenitors into S and G2/M phases (Fig. 3f). U73122, edelfosine, or CP-690550 also increased the population of apoptotic cells in the B cell progenitors in the presence of IL-7, compared with the IL-7-cultured control cells without any inhibitors (Fig. 3g). Therefore, inhibition of the PLCγ pathway impairs IL-7-mediated proliferation and survival of unexpanded B cell progenitors.

**PLCγ1/PLCγ2 double deficiency alters gene expression**. To further study the molecular mechanism underlying the defect of B cell development, we examined the global gene expression profile of PLCγ1/PLCγ2 double-deficient CLPs by RNA-seq analysis. CLPs (Lin$^-$IL-7R$^+$) were sorted from $Plcg1^{-/-}Plcg2^{-/-}$ and control mice, and subjected to RNA-seq analysis. Differential expression analysis demonstrated that $Plcg1^{-/-}Plcg2^{-/-}$, but not $Plcg1^{-/-}Plcg2^{+/+}$ or $Plcg1^{+/+}Plcg2^{-/-}$, CLPs displayed a marked change of the gene expression profile compared to $Plcg1^{+/+}Plcg2^{+/+}$ CLPs (Fig. 4a). Specifically, a total of 162 genes were significantly differentially expressed in $Plcg1^{-/-}Plcg2^{-/-}$ CLPs, including 97 upregulated genes and 65 downregulated genes (Fig. 4a). In contrast, a total of seven genes in $Plcg1^{-/-}Plcg2^{+/+}$ CLPs and a total of six genes in $Plcg1^{+/+}Plcg2^{-/-}$ CLPs were significantly differentially expressed (Fig. 4a). Ingenuity pathway analysis identified that certain lymphoid-lineage-affiliated genes were reduced in $Plcg1^{-/-}Plcg2^{-/-}$, relative to $Plcg1^{+/+}Plcg2^{+/+}$, CLPs (Fig. 4b). However, the expression of lymphoid specification factors, such as Runx1, E2A, EBF1, Tcf12, Notch1[51], was not significantly changed in $Plcg1^{-/-}Plcg2^{-/-}$, relative to $Plcg1^{+/+}Plcg2^{+/+}$, CLPs (RNA-seq data in the GEO database: GSE89352). Therefore, the PLCγ pathway does not control the lymphoid commitment but impairs further differentiation of CLPs.

Further, PLCγ1/PLCγ2 double-deficient B cell progenitors were subjected to global gene expression profiling. B cell progenitors (B220$^+$IgM$^-$IL-7R$^+$) were sorted from $Plcg1^{-/-}Plcg2^{-/-}$ and control mice and subjected to RNA-seq analysis. Differential expression analysis showed that $Plcg1^{-/-}Plcg2^{-/-}$, but not $Plcg1^{-/-}Plcg2^{+/+}$ or $Plcg1^{+/+}Plcg2^{-/-}$, B cell progenitors had a distinct gene expression profile compared to corresponding $Plcg1^{+/+}Plcg2^{+/+}$ B cell progenitors (Fig. 4c). Specifically, $Plcg1^{-/-}Plcg2^{-/-}$ B cell progenitors exhibited a total of 688 significantly altered genes, including 320 upregulated and 368 downregulated (Fig. 4c). By comparison, a total of 65 genes in $Plcg1^{-/-}Plcg2^{+/+}$ B cell progenitors and a total of 18 genes in $Plcg1^{+/+}Plcg2^{-/-}$ progenitors were significantly changed (Fig. 4c). Ingenuity pathway analysis showed that B cell lineage-affiliated and IL-7 signaling pathway-regulated genes were significantly downregulated in $Plcg1^{-/-}Plcg2^{-/-}$, relative to $Plcg1^{+/+}Plcg2^{+/+}$, B cell progenitors (Fig. 4d, e), consistent with the observation that PLCγ1/PLCγ2 double-deficient B cell progenitors failed to respond to IL-7 stimulation (Fig. 2a, b). The cell-cycle pathway was also strongly downregulated as many cell cycle-related genes were significantly reduced in $Plcg1^{-/-}Plcg2^{-/-}$ B cell progenitors (Fig. 4f). In addition, the expression of genes related to cell apoptosis was altered in $Plcg1^{-/-}Plcg2^{-/-}$, relative to $Plcg1^{+/+}Plcg2^{+/+}$, B cell progenitors (Fig. 4g), suggesting modest activation of "cell death" process (although this did not reach statistical significant). Of note, the expression of B cell lineage specification factors, such as E2A, EBF1 and Pax5[51], and Raptor and Rictor was not significantly altered in $Plcg1^{-/-}Plcg2^{-/-}$, relative to $Plcg1^{+/+}Plcg2^{+/+}$, B cell progenitors (RNA-seq data in the GEO database: GSE89352). A bar plot summarizing top activated and inhibited pathways in $Plcg1^{-/-}Plcg2^{-/-}$ pro-B cells was shown (Supplementary Fig. 7a). In addition, the reduction of some of the top listed genes was confirmed by quantitative real-time PCR (qRT-PCR) (Supplementary Fig. 7b). Taken together, these findings demonstrate that the PLCγ-dependent pathway does not control the B cell commitment/priming but directs gene expression to mediate IL-7-induced cell-cycle progression, differentiation and, possibly, survival during early B cell lymphopoiesis.

**Fig. 4** PLCγ1/PLCγ2 double deficiency alters gene expression in CLPs and B cell progenitors. BM from $Plcg1^{+/+}Plcg2^{+/+}$, $Plcg1^{-/-}Plcg2^{+/+}$, $Plcg1^{+/+}Plcg2^{-/-}$, or $Plcg1^{-/-}Plcg2^{-/-}$ mice were transplanted into lethally irradiated congenic wild-type CD45.1$^+$ mice. Six to eight weeks after transplantation, CLPs (CD45.2$^+$Lin$^-$IL-7R$^+$) and B cell progenitors (CD45.2$^+$B220$^+$IgM$^-$IL-7R$^+$) were sorted from the BM transplantation recipients and subjected to RNA-seq analysis. **a** Differentially expressed genes in $Plcg1^{-/-}Plcg2^{-/-}$ (upper), $Plcg1^{-/-}Plcg2^{+/+}$ (middle), or $Plcg1^{+/+}Plcg2^{-/-}$ (lower) relative to $Plcg1^{+/+}Plcg2^{+/+}$ CLPs. Differentially expressed genes with log2 fold-change ≥0.5 and FDR ≤0.05 are shown in red. **b** Heat maps of differentially expressed genes related to T and B lymphocytes in $Plcg1^{-/-}Plcg2^{-/-}$ relative to $Plcg1^{+/+}Plcg2^{+/+}$ CLPs. **c** Differentially expressed genes in $Plcg1^{-/-}Plcg2^{-/-}$ (upper), $Plcg1^{-/-}Plcg2^{+/+}$ (middle), or $Plcg1^{+/+}Plcg2^{-/-}$ (lower) relative to $Plcg1^{+/+}Plcg2^{+/+}$ B cell progenitors. Differentially expressed genes with log2 fold-change ≥0.5 and FDR ≤0.05 are shown in red. Heat maps of differentially expressed genes related to B cell lineage (**d**), IL-7 signaling (**e**), cell cycle (**f**), and cell apoptosis (**g**) in $Plcg1^{-/-}Plcg2^{-/-}$ relative to $Plcg1^{+/+}Plcg2^{+/+}$ B cell progenitors are shown. Data shown are obtained from three mice of each genotype

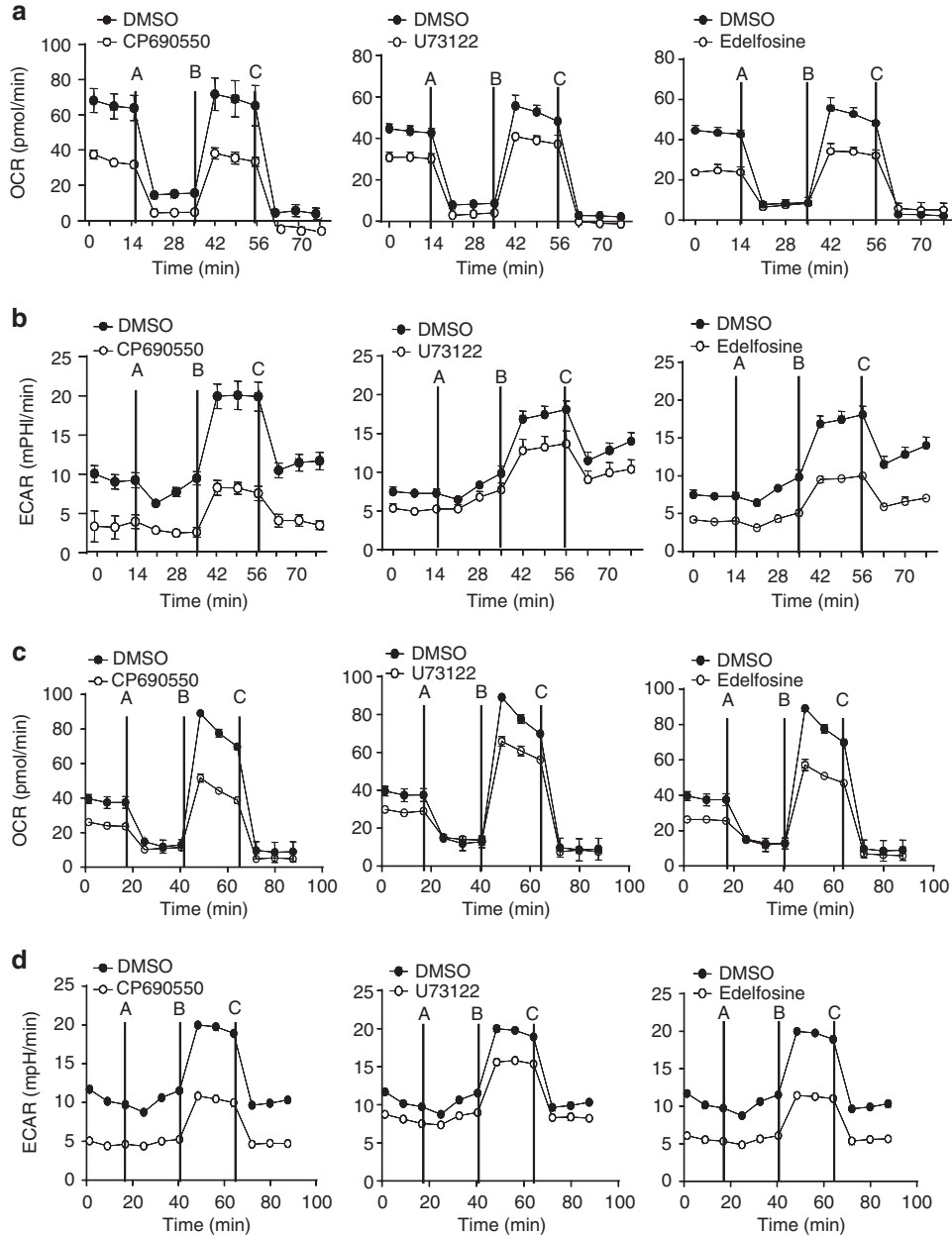

**Fig. 5** Inhibition of the PLCγ pathway impairs IL-7-mediated metabolism of B cell progenitors. **a**, **b** BM⁻derived Rag1-deficient B cell progenitors were cultured with IL-7 plus DMSO, CP690550, U73122, or edelfosine. **c**, **d** B220⁺IgM⁻IL-7R⁺ B cell progenitors were sorted from wild-type mice and cultured without IL-7 (medium) or with IL-7 plus DMSO, CP690550, U73122, or edelfosine. After measuring basal OCR (**a**, **c**) and ECAR (**b**, **d**), oligomycin A (A), FCCP (B), and antimycin (C) were sequentially added. Error bars show ± SEM. Data shown are obtained from 5 (**a**, **b**) and 3 (**c**, **d**) independent experimentsED: On your figures please ensure all instances of PLCγ refer to proteins, and <Emphasis Type="Italic">Plcg</Emphasis> is used to refer to the gene.

**The PLCγ pathway supports IL-7-mediated metabolism**. Cell proliferation is associated with cellular metabolism changes and the mechanistic target of rapamycin (mTOR) signaling pathway that controls cell metabolism[52]. We also examined the role of the PLCγ pathway in IL-7-mediated mitochondrial respiration by real-time measurement of the rate of oxygen consumption (OCR). Basal OCR level was established followed by sequential addition of the mitochondrial function modulators, oligomycin, FCCP, and antimycin A. U73122-treated or edelfosine-treated B cell progenitors exhibited a marked reduction of IL-7-mediated basal, uncoupled (proton leak), and maximal OCR compared to the control cells without inhibitor treatment (Fig. 5a). As controls, CP-690550 markedly reduced IL-7-mediated basal,

uncoupled, and maximal OCR (Fig. 5a). We next examined the role of the PLCγ pathway in IL-7-mediated glycolysis by real-time measurement of the rate of extracellular acidification (ECAR). Basal ECAR level was established following sequential addition of oligomycin, FCCP, and antimycin A. U73122 or edelfosine, similar to CP-690550, reduced IL-7-mediated basal, uncoupled, and maximal ECAR in B cell progenitors compared to the control cells without inhibitor treatment (Fig. 5b). Therefore, inhibition of the PLCγ pathway impairs IL-7-mediated metabolism of in vitro-expanded B cell progenitors.

Further, we examined the effect of inhibition of the PLCγ pathway on IL-7-mediated metabolism of unexpanded B cell progenitors. IL-7R⁺ B cell progenitors were sorted from wild-type

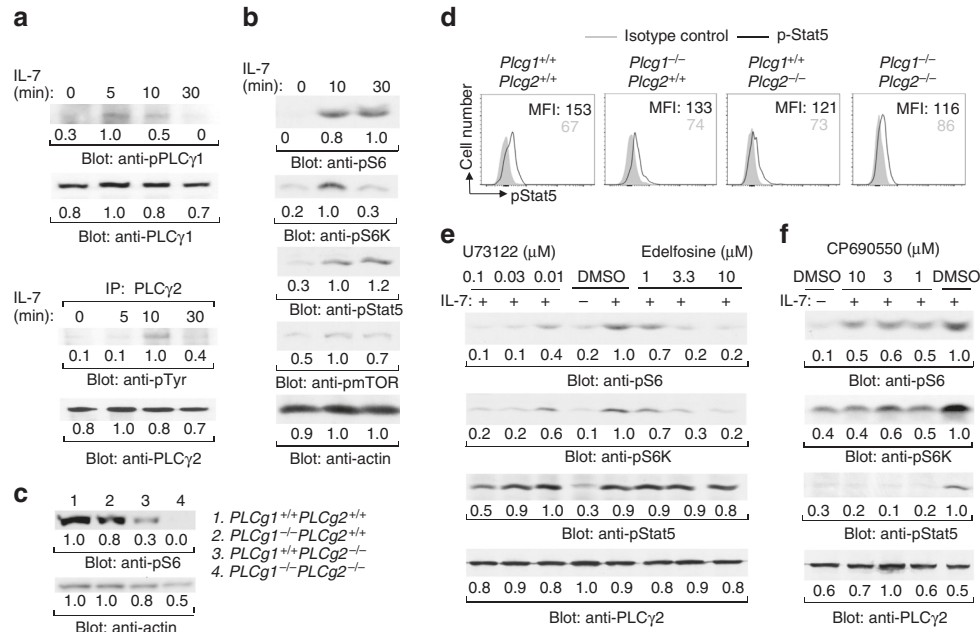

**Fig. 6** Inhibition of the PLCγ pathway impairs IL-7-induced mTOR activation. **a** PLCγ1 and PLCγ2 are phosphorylated upon IL-7 stimulation. Pro-B cells derived from *Rag1*-deficient mice were stimulated with IL-7. Cell lysates were subjected to western blot analysis (upper). Cell lysates were immunoprecipitated (IP) with anti-PLCγ2 antibodies and precipitated proteins were immunoblotted with the indicated antibodies (lower). **b** IL-7 stimulation activates the mTOR pathway. Pro-B cells derived from Rag1-deficient mice were stimulated with IL-7 and cell lysates were subjected to western blot analysis. **c** PLCγ1/PLCγ2 double deficiency impairs mTOR activation. YFP⁺CD45.2⁺ pre-pro-B cells (fraction A) were sorted from the indicated BM transplantation recipients and cell lysates were subjected to western blot analysis. **d** PLCγ1/PLCγ2 double deficiency does not affect Stat5 activation. BM cells from the indicated recipients were stained with anti-B220 and anti-CD43 antibodies, followed by intracellular staining with anti-phosphor-Stat5 (pStat5) antibodies or isotype controls. Numbers indicate the mean fluorescent intensity (MFI) of B220⁺CD43⁺ cells of pStat5 (black) or isotype controls (gray) staining. **e, f** Inhibition of the PLCγ pathway or Jak3 impairs IL-7-induced activation of mTOR. Pro-B cells derived from *Rag1*-deficient BM were pre-treated with DMSO, U73122, edelfosine, or CP-690550; and then stimulated with IL-7. Cell lysates were subjected to western blot analysis. The number beneath each band in the western blot indicates the relative intensity of the corresponding band. Data shown are representative of 3 (**a, b, d**), 4 (**c**), 5 (**e**), and 3 (**f**) independent experiments

mice, incubated with IL-7 plus DMSO or the inhibitors, and then subjected to OCR and ECAR measurement. U73122 or edelfosine markedly reduced IL-7-mediated basal and maximal OCR in unexpanded B cell progenitors compared to the control cells in the absence of any inhibitor (Fig. 5c). CP-690550 markedly reduced IL-7-mediated basal and maximal OCR in these B cell progenitors (Fig. 5c). Similar to CP-690550, U73122 or edelfosine reduced IL-7-mediated basal, uncoupled, and maximal ECAR in unexpanded B cell progenitors compared to the control cells without inhibitor treatment (Fig. 5d). In addition, manaolide reduced both IL-7-mediated basal, uncoupled, and maximal OCR and ECAR in unexpanded B cell progenitors (Supplementary Fig. 8). Therefore, inhibition of the PLCγ pathway impairs IL-7-mediated metabolism of B cell progenitors without in vitro expansion.

**The PLCγ pathway controls IL-7-induced mTOR activation.** To provide biochemical evidence demonstrating that the PLCγ pathway is involved in IL-7R signaling, we examined activation of PLCγ1 and PLCγ2 by IL-7 stimulation in B cell progenitors. *Rag1*-deficient pro-B cells lack the pre-BCR, excluding any potential interference of the pre-BCR with IL-7R signaling. Rag1-deficient pro-B cells express IL-7R, can be readily expanded in vitro, and thus are particularly suitable for the study of IL-7-mediated signaling and functions. Pro-B cells were derived from *Rag1*-deficient BM after co-culture with OP9 cells and IL-7. In these B cell progenitors, IL-7 stimulation activated both PLCγ1 and PLCγ2 as measured by protein tyrosine phosphorylation (Fig. 6a). In addition, IL-7 stimulation activated the two

downstream molecules, PKC and Ca²⁺, of the PLCγ pathway. Prolonged and constant IL-7 treatment elevated the level of intracellular free Ca²⁺, a consequence of IP₃ production (Supplementary Fig. 9a, b). Further, IL-7 stimulation activated phosphorylation of ribosomal protein S6, S6 kinase (S6K), and mTOR itself, indicators of mTOR activation (Fig. 6b). As expected, IL-7 stimulation activated tyrosine phosphorylation of Stat5 (Fig. 6b). Thus, in addition to Stat5, both PLCγ and mTOR are biochemically activated by IL-7 stimulation in B cell progenitors.

The mTOR signaling pathway controls cell metabolism. We found that the PLCγ pathway was important for the regulation of cellular metabolism (Fig. 5). These observations raise the question whether the PLCγ pathway is linked to mTOR activation. Therefore, we examined the role of PLCγ1 and PLCγ2 in mTOR activation in B cell progenitors. Owing to the scarcity of the remaining PLCγ1/PLCγ2 double-deficient B cell progenitors, we were only able to examine the basal level of S6 phosphorylation in these mutant progenitors. The pre-pro-B cells (YFP⁺CD45.2⁺B220⁺CD43⁺CD24⁻) were sorted from *Plcg1⁻/⁻Plcg2⁻/⁻* mice, and direct western blot analysis showed that the basal level of S6 phosphorylation was markedly decreased in *Plcg1⁻/⁻Plcg2⁻/⁻* relative to the corresponding *Plcg1⁺/⁺Plcg2⁺/⁺*, *Plcg1⁻/⁻Plcg2⁺/⁺*, or *Plcg1⁺/⁺Plcg2⁻/⁻* B cell progenitors (Fig. 6c). In contrast, phospho-flow cytometry showed that the basal level of Stat5 phosphorylation was comparable among *Plcg1⁺/⁺Plcg2⁺/⁺*, *Plcg1⁻/⁻Plcg2⁺/⁺*, *Plcg1⁺/⁺Plcg2⁻/⁻*, and *Plcg1⁻/⁻Plcg2⁻/⁻* pre-pro-B cells (Fig. 6d). These data show that PLCγ1/PLCγ2 double deficiency impairs mTOR activation in B cell progenitors.

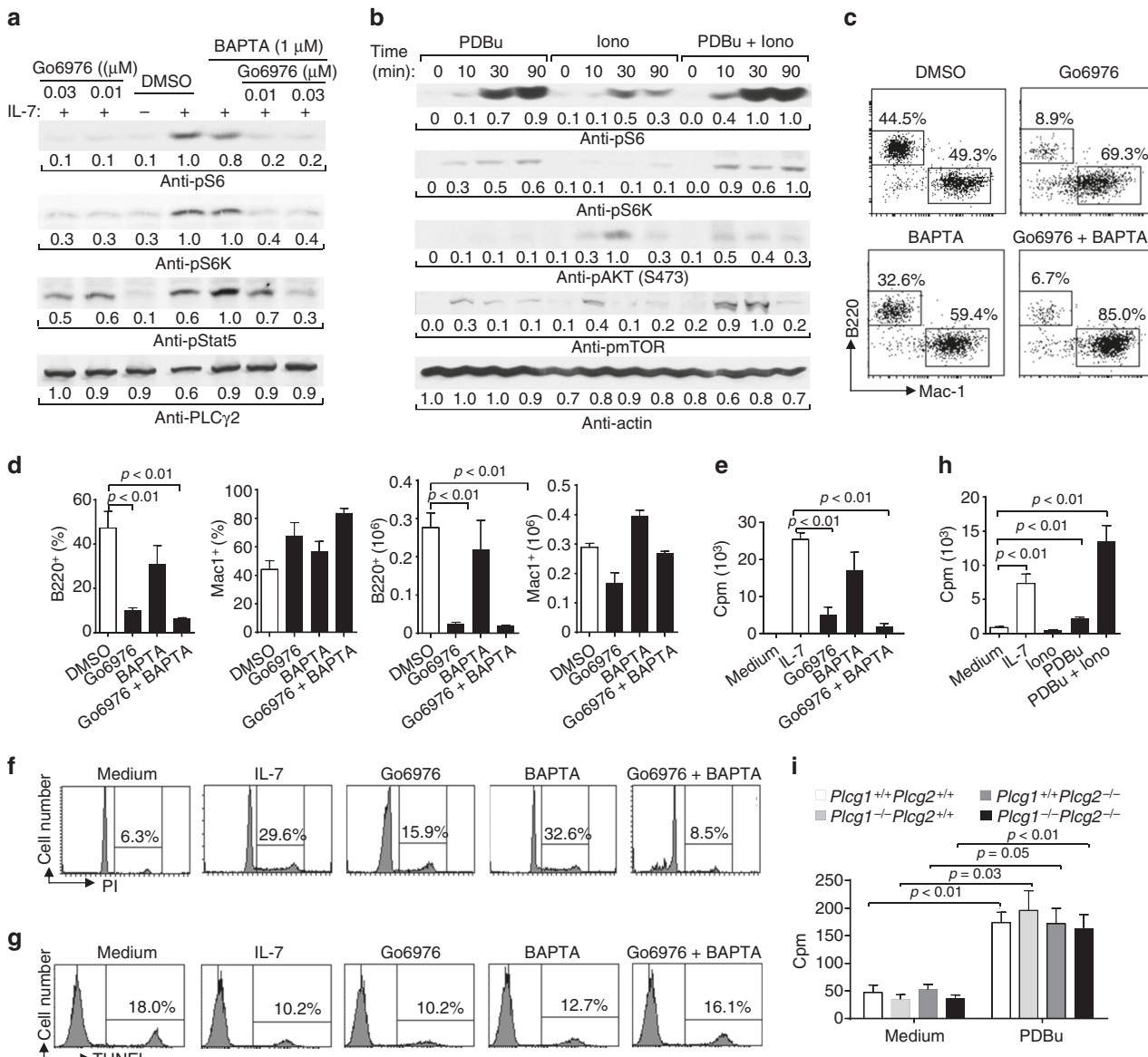

**Fig. 7** A major role of the DAG/PKC-dependent pathway in IL-7-mediated mTOR activation and functions. **a** Inhibition of the PKC pathway impairs IL-7-induced mTOR activation. Pro-B cells derived from Rag1-deficient BM were pre-treated with DMSO, Go6976, BAPTA, or Go6976 plus BAPTA, and stimulated with IL-7. Cell lysates were subjected to western blot analysis. **b** The PKC pathway activates mTOR in pro-B cells. Pro-B cells derived from Rag1-deficient BM were starved and stimulated with PDBu, ionomycin, or both. Cell lysates were subjected to western blot analysis. **c**, **d** Inhibition of the PKC pathway impairs IL-7-dependent B cell production. Lin⁻ BM cells from wild-type mice were cultured on OP9 cells plus IL-7 for 1 day and then were treated with DMSO, Go6976, or Go6976 plus BAPTA. Numbers indicate percentages of B220⁺ and Mac-1⁺ cells in the gated live cells (**c**). Bar graphs show the percentages and numbers of B220⁺ and Mac-1⁺ cells (**d**). **e–g** Inhibition of the PKC pathway impairs IL-7-mediated B cell progenitor proliferation but not survival. Pro-B cells derived from Rag1-deficient BM were cultured without IL-7 (medium) or with IL-7 plus DMSO, Go6976 or Go6976, and BAPTA. Cell proliferation was determined by [³H] thymidine incorporation (**e**), cell-cycle analysis was performed by PI staining (**f**), and cell survival was measured by TUNEL staining (**g**). **h** PKC pathway activation induces B cell progenitor proliferation. Pro-B cells derived from Rag1-deficient BM were cultured without IL-7 (medium) or with IL-7, ionomycin (Iono), PDBu, or PDBu plus iono. Cell proliferation was determined by [³H] thymidine incorporation. **i** YFP⁺CD45.2⁺B220⁺CD43⁺CD24⁺ B cell progenitors were sorted from the congenic wild-type CD45.1⁺ recipients received $Plcg1^{+/+}Plcg2^{+/+}$, $Plcg1^{-/-}Plcg2^{+/+}$, $Plcg1^{+/+}Plcg2^{-/-}$, or $Plcg1^{-/-}Plcg2^{-/-}$ BM, and cultured with or without PDBu. Cell proliferation was determined by [³H] thymidine incorporation. The number beneath each band in the western blot indicates the relative intensity of the corresponding band. Error bars show ± SEM. Data shown are obtained from or representative of 3 (**a–d**, **f**, **g**), 4 (**i**), and 6 (**e**, **h**) independent experiments

To further demonstrate that the PLCγ pathway regulates mTOR activation by IL-7 stimulation, we examined the effect of inhibition of this pathway on IL-7-induced activation of mTOR in B cell progenitors without the pre-BCR. Pro-B cells were derived from *Rag1*-deficient BM cells. Pre-treatment of these progenitors with U73122 reduced IL-7-induced phosphorylation of S6 and S6K compared with the control without inhibitor pre-

treatment (Fig. 6e). Of note, U73122 pre-treatment had no effect on IL-7-induced phosphorylation of Stat5 in these B cell progenitors (Fig. 6e). As a control, CP-690550 reduced IL-7-induced phosphorylation of S6, S6K, and Stat5 in these B cell progenitors (Fig. 6f). In addition, manoalide reduced IL-7-induced phosphorylation of S6 compared to the control without inhibitor pre-treatment (Supplementary Fig. 10a). In contrast,

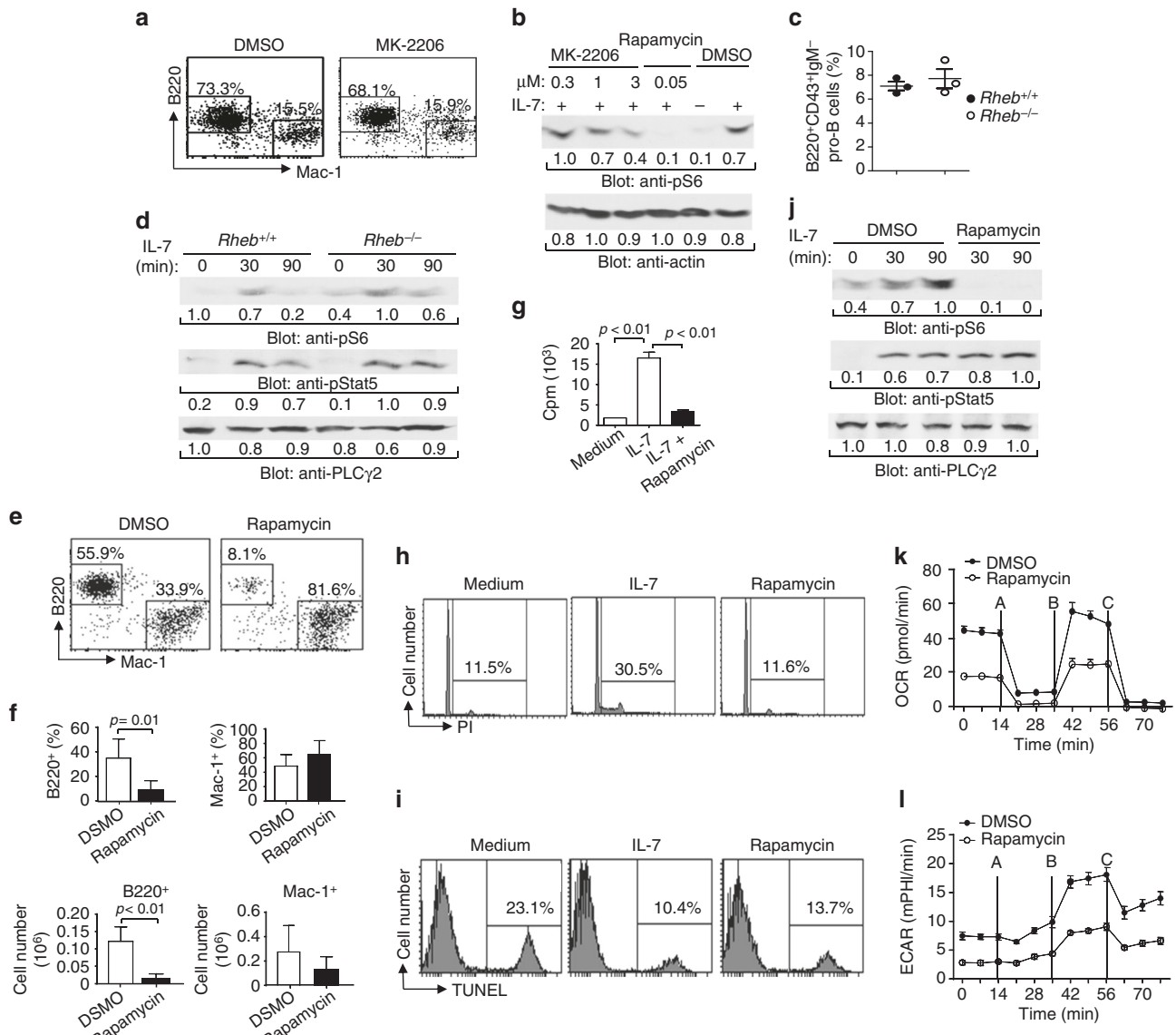

**Fig. 8** IL-7-mediated mTOR activation is Akt/Tsc/Rheb pathway-independent. **a** AKT inhibition does not affect IL-7-dependent B cell production. Wild-type Lin⁻ BM cells were cultured on OP9 cells plus IL-7 for 1 day and then treated with DMSO or MK-2206. Numbers indicate percentages of B220⁺ and Mac-1⁺ cells in the gated live cells. **b** AKT inhibition does not block S6 phosphorylation. Pro-B cells derived from *Rag1*-deficient BM were pre-treated with DMSO, MK-2206 or rapamycin for 10 min and then stimulated with IL-7 for 90 min. Cell lysates were subjected to Western analysis. **c** Rheb deficiency does not affect early B cell development. The percentages of pro-B cells in the gated BM B220⁺ cells of *Rheb*⁺/⁺ and *Rheb*⁻/⁻ mice. **d** IL-7-induced mTOR activation is Rheb-independent. Pro-B cells derived from *Rheb*⁺/⁺ and *Rheb*⁻/⁻ BM were stimulated with IL-7. Cell lysates were subjected to western analysis. **e**, **f** mTOR pathway inhibition impairs IL-7-dependent B cell production. Wild-type Lin⁻ BM cells were cultured on OP9 cells plus IL-7 for 1 day and then treated with DMSO or rapamycin. Numbers indicate percentages of B220⁺ and Mac-1⁺ cells in the gated live cells (**e**). Bar graphs show the percentages and numbers of B220⁺ and Mac-1⁺ cells (**f**). **g**–**i** mTOR pathway inhibition blocks or partially impairs IL-7-mediated B cell progenitor proliferation or survival, respectively. Pro-B cells derived from Rag1-deficient BM were cultured without IL-7 (medium) or with IL-7 plus DMSO or rapamycin. Cell proliferation was determined by [³H] thymidine incorporation (**g**), cell-cycle analysis was performed by PI staining **h**, and cell survival was measured by TUNEL staining (**i**). **j**–**l** mTOR inhibition impairs IL-7-mediated S6 phosphorylation and metabolism in B cell progenitors. Pro-B cells derived from Rag1-deficient BM were stimulated with IL-7 plus or minus rapamycin. Cell lysates were subjected to western analysis (**j**). These pro-B cells were cultured with IL-7 plus DMSO or rapamycin. After basal OCR (**k**) and ECAR (**l**) measurement, oligomycin A (A), FCCP (B), and antimycin (C) were sequentially added. Error bars show ± SEM. Data shown are obtained from or representative of 2 (**a**, **b**, **d**), 3 (**c**), 5 (**e**, **f**, **k**, **l**), 6 (**g**, **j**), and 7 (**h**, **i**) independent experiments

manoalide treatment had no effect on IL-7-induced phosphorylation of Stat5 in B cell progenitors (Supplementary Fig. 10a). Of note, U73122 or edelfosine inhibited BCR-induced Ca²⁺ flux in splenic mature B cells (Supplementary Fig. 10b), demonstrating these inhibitors could directly blocked the activity of PLCγs. Thus, inhibition of the PLCγ pathway impairs IL-7-induced mTOR but not Stat5 activation, demonstrating that the PLCγ

pathway functions upstream of mTOR activation during IL-7 signaling.

**The DAG/PKC pathway controls IL-7-induced mTOR activation.** To further confirm that the PLCγ pathway functions upstream of IL-7-induced mTOR activation, we examined the

roles of its two downstream cascades, the DAG/PKC-dependent and IP$_3$/Ca$^{2+}$-dependent pathways, in IL-7-mediated mTOR activation in B cell progenitors. Pro-B cells derived from Rag1-deficient BM cells were pre-treated with the PKC inhibitor Go6976 or/and the membrane permeable Ca$^{2+}$ chelator BAPTA. Go6976 or Go6976 plus BAPTA but not BAPTA alone completely inhibited IL-7-induced phosphorylation of S6 and S6K compared with the control without an inhibitor (Fig. 7a). However, Go6976 or Go6976 plus BAPTA had little effect on IL-7-induced phosphorylation of Stat5 in these B cell progenitors (Fig. 7a). Thus, inhibition of the PKC but not the Ca$^{2+}$ pathway impairs mTOR activation by IL-7.

To further investigate the roles of the DAG/PKC-dependent and IP$_3$/Ca$^{2+}$-dependent pathways in mTOR activation, we examined the effect of the PKC agonist phorbol 12,13-dibutyrate (PDBu) and the Ca$^{2+}$ ionophore ionomycin on the phosphorylation of S6 and S6K in B cell progenitors. Pro-B cells derived from Rag1-deficient BM cells were stimulated with PDBu or/and ionomycin. PDBu or PDBu plus ionomycin markedly and ionomycin alone slightly induced phosphorylation of S6 and S6K (Fig. 7b). In addition, PDBu, PDBu plus ionomycin or ionomycin alone induced phosphorylation of mTOR itself (Fig. 7b). Of note, PDBu barely and ionomycin or PDBu plus ionomycin slightly induced Akt phosphorylation at S473 (Fig. 7b), whereas PDBu, ionomycin, or PDBu plus ionomycin barely induced Akt phosphorylation at T308 (Supplementary Fig. 11), indicating that PDBu might activate the mTOR pathway independently of Akt. Thus, the PKC pathway by itself can largely activate mTOR, whereas the Ca$^{2+}$ pathway may contribute to mTOR activation in B cell progenitors.

**The DAG/PKC pathway directs IL-7-induced cell proliferation.** To further investigate the roles of the DAG/PKC-dependent and IP$_3$/Ca$^{2+}$-dependent pathways in IL-7-mediated function, we examined the effect of inhibition of one or both of these two downstream cascades on IL-7-mediated functions in B cell progenitors. We found that Go6976 or Go6976 plus BAPTA blocked the development of wild-type BM progenitors into B cells but not Mac1$^+$ myeloid cells following IL-7-mediated co-culture on OP9 cells (Fig. 7c, d). In contrast, BAPTA itself had little effect on IL-7-mediated B cell development in vitro (Fig. 7c, d).

Further, the cultured cells were subjected to staining with a combination of anti-B220, anti-CD43, anti-BP-1, and anti-CD24. Go6976 or Go6976 plus BAPTA but not BAPTA itself markedly increased the percentages of pre-pro-B cells (fraction A) and markedly decreased the percentages of late pro-B/early pre-B cells (fraction C/C′) (Supplementary Fig. 12a, b). In addition, Go6976 or Go6976 plus BAPTA but not BAPTA itself reduced the numbers of pre-pro-B cells (fraction A), markedly decreased the numbers of early pro-B (fraction B) and completely diminished the numbers of late pro-B/early pre-B cells (fraction C/C′) (Supplementary Fig. 12c). Thus, inhibition of PKC, but not chelation of Ca$^{2+}$, prevents IL-7-mediated B cell production in vitro mainly at the pre-pro-B stage.

Consistently, Go6976 but not BAPTA markedly reduced the rate of IL-7-induced $^3$H-thymidine incorporation of OP9 co-culture-derived Rag1-deficient B cell progenitors (Fig. 7e). Of note, Go6976 plus BAPTA further decreased the rate of IL-7-induced $^3$H-thymidine incorporation of B cell progenitors (Fig. 7e). PI staining analysis found that IL-7 promoted the entry of B cell progenitors derived from OP9 co-culture into S and G2/M phases and Go6976, but not BAPTA, markedly inhibited IL-7-induced cell-cycle progress of these B cell progenitors (Fig. 7f). Again, Go6976 plus BAPTA further blocked IL-7-induced cell-cycle progress of these B cell progenitors (Fig. 7f). However, the

TUNEL assay revealed that Go6976 or BAPTA had no effect on IL-7-mediated survival of B cell progenitors compared with the control cells without inhibitor treatment (Fig. 7g). Moreover, Go6976, but not BAPTA, blocked IL-7-induced cell-cycle progression of unexpanded B cell progenitors (Supplementary Fig. 13a). Go6976 or BAPTA inhibited IL-7-mediated survival of unexpanded B cell progenitors compared to vesicle-treated control cells (Supplementary Fig. 13b). Thus, inhibition of PKC, but not chelation of Ca$^{2+}$, markedly impedes, whereas inhibition of both pathways completely blocks IL-7-induced proliferation of expanded B cell progenitors. Inhibition of either pathway has no effect on IL-7-mediated survival of expanded B cell progenitors. However, inhibition of PKC, but not chelation of Ca$^{2+}$, blocks IL-7-induced proliferation of unexpanded B cell progenitors, whereas inhibition of either pathway impairs IL-7-mediated survival of unexpanded B cell progenitors.

Moreover, the PKC agonist PDBu but not the Ca$^{2+}$ ionophore ionomycin clearly induced $^3$H-thymidine incorporation of BM-derived Rag1-deficient pro-B cells (Fig. 7h). PDBu and ionomycin together markedly induced $^3$H-thymidine incorporation of the pro-B cells (Fig. 7h). Importantly, PDBu induced $^3$H-thymidine incorporation in PLCγ1/PLCγ2 double-deficient B cell progenitors to the same extent as it did in control cells (Fig. 7i). Thus, activation of the PKC-dependent but not the Ca$^{2+}$-dependent pathway can induce proliferation of B cell progenitors, whereas activation of both pathways further enhances B cell progenitor proliferation. Activation of the PKC-dependent pathway can overcome PLCγ1/2 double deficiency to induce cell proliferation independent of IL-7. Taken together, these data demonstrate that the PKC-dependent pathway has a major role in IL-7-mediated development of B progenitors.

**PKC-dependent mTOR activation in early B cell development.** Many receptors normally activate mTOR through the PI3K/Akt/Tsc/Rheb pathway[53]. Rheb (Ras homolog enriched in brain) is a GTPase that activates mTOR. The tuberous sclerosis 1 (Tsc1) and Tsc2 complex functions as a GTPase activating protein (GAP) that directly inhibits the Rheb activation. Typically, activated Akt phosphorylates Tsc2, resulting in the dissociation and functional inactivation of the Tsc1/2 complex and subsequent Rheb/mTOR activation. However, we found that PDBu induced mTOR activation without Akt activation in pro-B cells (Fig. 7b), suggesting that IL-7/PKC might activate the mTOR pathway independently of the Akt/Tsc/Rheb pathway. Thus, we examined the effect of the AKT inhibitor MK-2206 on IL-7-induced B cell production from Lin$^-$ BM progenitors in vitro. In the presence of MK-2206, wild-type BM progenitors mainly developed into B cells with some myeloid cells following IL-7-mediated co-culture with OP9 cells, similar to control progenitors in the absence of the inhibitor (Fig. 8a). Thus, inhibition of AKT fails to block IL-7-dependent B cell production.

Further, we examined the effect of MK-2206 on IL-7-mediated mTOR activation. Two relatively lower concentrations of MK-2206 failed to block IL-7-induced S6 phosphorylation in pro-B cells derived from Rag1-deficient BM (Fig. 8b), but markedly impaired BCR-mediated mTOR activation in splenic B cells (Supplementary Fig. 14). In contrast, rapamycin blocked IL-7-induced S6 phosphorylation (Fig. 8b). Thus, inhibition of AKT fails to block IL-7-induced mTOR activation.

To further examine the role of IL-7-mediated mTOR activation by the Akt/Tsc/Rheb pathway, we examined the effect of Rheb deficiency on B cell development. In BM, the populations of total B220$^+$ B cells as well as pro- (B220$^+$CD43$^+$), pro/pre- (B220$^+$IgM$^-$), immature (B220$^+$IgM$^+$), and mature (B220$^{hi}$IgM$^+$) B cells were normal in Rheb-deficient relative to wild-type mice (Fig. 8c;

Supplementary Fig. 15a, b). In addition, within the B220[+]CD43[+] cells, Rheb-deficient relative to wild-type mice displayed slightly decreased percentages of pre-pro-B (fraction A), normal percentages of early pro-B (fraction B), and increased percentages of late pro-B/early pre-B (fraction C/C′) cells (Supplementary Fig. 15c). Compared to wild-type mice, Rheb-deficient mice had normal numbers of pre-pro-B and pro-B and increased numbers of late pro-B/early pre-B cells (Supplementary Fig. 15d). Thus, Rheb deficiency does not affect IL-7-mediated pre-pro-B to early pro-B transition.

We also examined the effect of Rheb deficiency on IL-7-mediated mTOR activation. Pro-B cells were derived from Rheb-deficient BM cells through co-culture with OP9 cells. IL-7-induced phosphorylation of S6 was comparable between Rheb-deficient and wild-type pro-B cells (Fig. 8d). In contrast, BCR-induced phosphorylation of S6 was abolished in Rheb-deficient relative to wild-type splenic B cells (Supplementary Fig. 16). These data demonstrate that Rheb is required for BCR-induced mTOR activation in peripheral B cells, but not required for IL-7-induced mTOR activation in B cell progenitors. Thus, IL-7 activates mTOR through the PLCγ/PKC pathway, independently of the conventional Akt/Tsc/Rheb pathway. The Akt/Tsc/Rheb pathway is not required for IL-7R-mediated early B cell development.

We investigated whether mTOR activation by IL-7 through the PLCγ pathway is critical for early B cell development. To this end, we examined the effect of the mTOR inhibitor rapamycin on IL-7-mediated functions. Rapamycin inhibited wild-type Lin[−] BM progenitors from developing into B but not myeloid cells in response to IL-7 on OP9 cells (Fig. 8e, f). Rapamycin also blocked IL-7-induced [3]H-thymidine incorporation of B cell progenitors derived from BM as compared with the control cells in the absence of the inhibitor (Fig. 8g). Further, PI staining analysis demonstrated that rapamycin blocked IL-7-induced entry of these B cell progenitors into S and G2/M phases (Fig. 8h). In contrast, the TUNEL assay revealed that rapamycin barely increased the population of apoptotic cells in IL-7-cultured B cell progenitors as compared to the control cells without the inhibitor (Fig. 8i). Moreover, rapamycin blocked IL-7-induced S6 phosphorylation, but not Stat5 phosphorylation in pro-B cells derived from Rag1-deficient BM (Fig. 8j). Finally, rapamycin markedly reduced IL-7-mediated basal, uncoupled, and maximal OCR and ECAR (Fig. 8k, l). These data demonstrate that inhibition of mTOR activation blocks IL-7-mediated B cell production, proliferation, and metabolism. Taken together, we conclude that PLCγ/PKC but not Akt/Tsc/Rheb pathway-dependent activation of mTOR is essential for IL-7-mediated early B cell development.

## Discussion

It is known that PLCγ1 and PLCγ2 have distinct functions and have an important role in TCR and BCR-mediated T and B cell maturation and function, respectively[37, 43]. However, due to a potential redundant role of the two PLCγs, the biological function of this important pathway is not fully understood. For example, the role of the PLCγ pathway in regulating the biological functions of cytokines is unknown, although erythropoietin (Epo) or macrophage colony-stimulating factor (M-CSF) induce tyrosine phosphorylation of both PLCγ1 and PLCγ2[54, 55]. Our current study for the first time demonstrates that PLCγ1 and PLCγ2 have overlapping functions and both participate in IL-7 signaling, revealing a critical but as yet unknown role of the PLCγ pathway in IL-7-mediated early lymphopoiesis. Signals emanating from the IL-7R control the differentiation of CLPs into pro-B cells, the proliferation and survival of pro-B cells, the rearrangement of IgH gene, and the expansion of pre-B cells[6, 8, 9, 47, 56]. The Jak1/

Jak3/Stat5 pathway has a critical role in controlling these IL-7-mediated functions[21–23]. Importantly, our findings demonstrate that the PLCγ pathway directs IL-7-controlled early B cell development independently of the Stat5 pathway. However, a previous study has shown that a constitutively active Stat5 can partially restore pro-B cell expansion in IL-7R-deficient mice[57], whereas our current data indicate that IL-7-induced activation of Stat5 is not sufficient to drive early B cell development in the absence of PLCγ activation. This discrepancy could be due to many differences, such as amplitude and kinetics, between IL-7-induced Stat5 activation and constitutive Stat5 activation. Taken together, all evidence supports the notion that the Stat5 pathway and the PLCγ pathway are individually necessary but not sufficient to fully drive IL-7-mediated early B cell development.

In addition to the IL-7R, the pre-BCR also controls early B cell lymphopoiesis[3, 47]. Given the fact that both PLCγs are important for antigen receptor-mediated function[37, 43], it is possible that both PLCγ1 and PLCγ2 are required for pre-BCR-mediated expansion of pre-B cells[1–3]. Disruption of pre-BCR function arrests B cell development at the late pro- and pre-B cell (fraction C/C′) stage[5]. However, we found that PLCγ1/PLCγ2 double deficiency, similar to that of IL-7 or IL-7R, blocked B cell development at the early pre-pro-B (fraction A) stage, reduced the numbers of all subsets of DN thymocytes, and decreased the numbers of CLPs[7, 48–50, 58]. The similarities between the effect of PLCγ1/PLCγ2 double deficiency and that of IL-7 or IL-7R deficiency on early B and T cell development supports the notion that both PLCγ1 and PLCγ2 participate in IL-7-mediated function. We find that PLCγ1/PLCγ2 double deficiency disrupts IL-7 function, arresting B cell development at the pre-pro-B cell stage and thus leading to the loss of pro-B cells. Rag1 deficiency disrupts pre-BCR expression and prohibits pre-BCR-mediated expansion of pre-B cells, arresting B cell development at the pro-B cell stage. Both pre-pro- and pro-B cells respond to the IL-7 stimulation, although their responses might have subtle differences. Nonetheless, Rag1-deficient pro-B cells express IL-7R and are especially suitable for the biochemical study of IL-7-mediated signaling. In vitro OP9 co-culture and biochemical studies further demonstrate that PLCγ1 and PLCγ2 participate in IL-7R signaling and are critical for IL-7-mediated B cell proliferation and production. Nonetheless, it is possible that PLCγ1 and PLCγ2 are involved in pre-BCR-mediated functions. Although single deficiency of PLCγ1 or PLCγ2 has no obvious effect on early B cell development[37, 43], both PLCγs might have a redundant and critical role in pre-BCR-mediated functions. Further studies of mice with PLCγ1/PLCγ2 double deficiency in pre-B cells are warranted.

We show that the PLCγ pathway is critical for IL-7-induced cell proliferation. PLCγ1/PLCγ2 double deficiency or inhibition of PLCγ impaired IL-7-induced cell proliferation in primary and in vitro-expanded B cell progenitors. Although IL-7 deficiency arrests B cell development largely at the pre-pro- to pro-B cell transition, CLPs, pre-pro-B, and pro-B cells are all responsive to IL-7[6, 8, 9]. PLCγ1/PLCγ2 double deficiency arrested B cell development at the pre-pro-B cell stage. Notably, the remaining residual PLCγ1/PLCγ2 double-deficient pro-B cells were defective in response to IL-7. RNA-seq analysis demonstrated that PLCγ1/PLCγ2 double-deficient B cell progenitors displayed a significantly reduced expression of many cell cycle-related genes, indicating a strongly downregulated cell-cycle pathway in the mutant B cell progenitors. In contrast, the role of the PLCγ pathway in IL-7-mediated cell survival is less clear. Inhibition of PLCγ increased cell apoptosis in unexpanded B cell progenitors, but not in the cells following in vitro expansion. TUNEL assay failed to detect a significant increase of apoptosis in PLCγ1/PLCγ2 double-deficient B cell progenitors. However, RNA-seq

analysis showed that the "cell death" process trended higher in PLCγ1/PLCγ2 double-deficient B cell progenitors. Thus, the PLCγ pathway controls IL-7-mediated proliferation and, possibly, survival of B cell progenitors. Previous studies have demonstrated that providing a survival signal is critical for IL-7R-mediated early T cell development[59, 60]. It is likely that the PLCγ-dependent pathway controls both IL-7-mediated proliferation and survival of T cell progenitors.

Pro-B cells are highly responsive to IL-7 stimulation[6, 7, 10, 11]. Rag deficiency arrests B cell development at the pro-B cell stage. Rag-deficient pro-B cells express IL-7R, can be readily expanded in vitro, and are particularly suitable for the study of IL-7-mediated signaling and functions. Inhibition of the PLCγ pathway reduced IL-7-mediated oxygen consumption (oxidative phosphorylation) and extracellular acidification (glycolysis) in unexpanded or in vitro-expanded B cell progenitors, demonstrating a critical role of the PLCγ pathway in controlling IL-7-regulated cellular metabolism. Cell proliferation is supported by cellular metabolism, which is controlled by the mTOR signaling pathway[52, 53, 61]. Importantly, we discovered that the PLCγ pathway-regulated IL-7-induced activation of mTOR. Typically, activation of mTOR depends on the PI3K/Akt/Tsc/Rheb pathway[53, 61, 62]. Activation of the lipid kinase PI3K leads to production of $PIP_3$ that recruits Akt to the plasma membrane, resulting in Akt activation through its dual phosphorylation by PDK1 and mTORC2. Activated Akt phosphorylates Tsc2, leading to functional inactivation of the Rheb GAP Tsc1/2 complex, and subsequent elevation of GTP-bound Rheb and activation of mTOR. However, Rheb deficiency had no effect on IL-7-induced mTOR activation in B cell progenitors, but disrupted BCR-induced mTOR activation in peripheral B cells, demonstrating that IL-7 uniquely activates mTOR through the PLCγ pathway independently of the conventional Akt/Tsc/Rheb pathway. In addition, inhibition of AKT blocks BCR-induced but not IL-7-induced mTOR activation. Consistently, single or double deficiency of *Akt1* and *Akt2* has no effect on IL-7-mediated early B cell development[63]. However, deficiency of *PDK1* inhibits IL-7-mediated proliferation of B cell progenitors, which could be due to PDK1 targeting Akt and other signaling molecules, such as PKC[64]. PLCγ hydrolyzes $PIP_2$ to generate DAG and $IP_3$ that induces PKC activation and intracellular $Ca^{2+}$ release, respectively[26, 27]. Inhibition of PKC but not chelation of intracellular calcium markedly decreased IL-7-induced mTOR activation. In addition, PKC agonist but not calcium ionophore markedly activated mTOR. Our data demonstrate that PKC, but not $Ca^{2+}$-dependent downstream pathway of PLCγ has a central role in activating mTOR. Of note, PKC agonist activated mTOR without Akt activation in pro-B cells, supporting the notion that the PLCγ pathway activates mTOR independently of the Akt/Tsc/Rheb pathway. Consistently, previous studies have shown that PKC is able to activate mTOR in mouse epithelial cells and glioma independently of Akt[65, 66]. Of note, the $Ca^{2+}$-dependent pathway is required for classical PKC activation[58], and thus this pathway may also contribute to mTOR activation through PKC. In fact, we found here that PKC agonist plus calcium further induced mTOR activation. These results demonstrate the PKC-dependent downstream pathway of PLCγ plays a major role in mTOR activation, whereas the $Ca^{2+}$-dependent downstream pathway contributes to it. The importance of the PLCγ/PKC/mTOR pathway in controlling early B cell development is supported by studies of mice with genetic disruption of this pathway via Raptor deletion[67, 68]. Raptor deficiency impairs early B cell development, leading to a marked reduction in peripheral B cells[67, 68]. Consistent with our current findings, B cell-specific Raptor deletion blocks early B cell development at the pre-pro-B cell stage, resulting in a decrease in early pro-B (fraction B)[67].

PLCγ1/PLCγ2 double, but not single, deficiency markedly altered the gene expression profile in CLPs. Lymphoid-lineage-affiliated genes, but not lymphoid specification factors, such as Runx1, E2A, EBF1, Tcf12, and Notch1[51], were identified as significantly reduced genes in PLCγ1/PLCγ2 double-deficient CLPs. Thus, the PLCγ pathway is not required for lymphoid commitment of CLPs but is important for their further differentiation. Previous studies have shown that IL-7 controls lymphoid commitment of CLPs through inducing EBF expression[50]. It is quite possible that IL-7 directs lymphoid commitment of CLPs through a PLCγ-independent pathway. Similarly, PLCγ1/PLCγ2 double-deficient but not single-deficient B cell progenitors exhibited a distinct gene expression profile compared to corresponding wild-type cells. B cell-affiliated, IL-7 signaling pathway-regulated, and cell cycle-related genes were significantly reduced in PLCγ1/PLCγ2 double-deficient B cell progenitors. However, the expression of B cell lineage specification factors, such as E2A, EBF1, and Pax5[51], was not significantly altered in PLCγ1/PLCγ2 double-deficient B cell progenitors. Thus, the PLCγ-dependent pathway seems not to control B cell commitment/priming, but directs IL-7-induced cell-cycle progression and differentiation. Previous studies have demonstrated that IL-7 controls B cell commitment/priming through Stat5-dependent upregulation of EBF[11]. Of note, the expression of genes related to Stat5 signaling was not reduced in PLCγ1/PLCγ2 double-deficient B cell progenitors, consistent with our observation that PLCγ1/PLCγ2 double deficiency did not affect IL-7-induced Stat5 activation. All evidences support the notion that the Stat5 pathway and the PLCγ pathway have a critical and distinct role in IL-7-mediated early B cell development. The Stat5/EBF axis is critical for IL-7-driven B cell lineage commitment/priming of B cell progenitors, whereas the PLCγ/mTOR axis directs IL-7-induced cell-cycle progression, differentiation and, possibly, survival during early B cell lymphopoiesis. Our findings indicate that disruption of the PLCγ pathway might lead to a severe combined immunodeficiency with absent T cells and B cells in humans.

## Methods

**Mice and reagents.** *Plcg1^flox* and *Plcg2^−/−* mice were described previously[37, 43]. Mx-1Cre transgenic and *Rag1^−/−* mice were purchased from Jackson Laboratory. CD45.1 mice were purchased from Jackson Laboratory or Charles River Laboratories. Rosa-26-YFP mice were generously provided by Dr. F. Costantini (Columbia University, New York). RosaCreERT2 mice were generously provided by Dr. Thomas Ludwig (Columbia University, New York). The experimental and control mice were on a mixed 129xC57BL/6 background and 8–12-weeks old, and include both males and females. Animal protocols were approved by the Medical College of Wisconsin Institutional Animal Care and Use Committee.

IL-7 and Flt3 ligand were from R&D Systems. SCF was purchased from Peprotech. 4-Hydroxytamoxifen (H7904) was from Sigma. U73122 (662035), rapamycin (553210), BAPTA/AM (196419), and Go6976 (365250) were purchased from CalBiochem EMD Millipore. CP-690550 citrate (4556) and edelfosine (3022) were purchased from TOCRIS Bioscience. MK-2206 (S1078) was purchased from Sellechem. Manoalide (ab141554) was purchased from Abcam. In all the in vitro inhibitor experiments, 0.3 μM of U73122, 3.3 μM of edelfosine, 0.05 μM of maoalide, 1 μM of CP-690550, 0.03 μM of Go6976, 0.05 μM of manoalide, 1 μM of BAPTA, and 50 nM of rapamycin were used unless stated otherwise. For ex vivo inhibitor experiments, 0.05 μM of U73122, 10 μM of edelfosine, 0.05 μM of manoalide, 1 μM of CP-690550, 0.15 μM of Go6976, 0.01 μM of BAPTA, and 50 nM of rapamycin were used.

**BM transplantation.** Experimental and control mice at 6–12 weeks of age were intraperitoneally injected with PolyI:C (GE Healthcare Life Sciences) every other day for three times and then killed 3 weeks after the last injection for BM isolation. CD45.1$^+$ mice were irradiated with 1000 rads and then injected with $2 \times 10^6$ total BM cells from control and experimental mice via tail vein. Mice were analyzed 6–8 weeks after BM transplantation.

**Flow cytometry and cell sorting.** Single-cell suspension from the BM of the experimental and control mice were treated with Gey's solution to lyse red blood cells (RBCs) and resuspended in PBS with 2% FBS. The cells were stained with a combination of fluorescence-conjugated antibodies. PE-conjugated anti-B220

(RA3-6B2), anti-CD11b (M1/70), anti-CD4 (GK1.5), anti-Gr.1 (RB6.8C5), anti-Ter119 (Ter119), and anti-IL-7R (A7R34), Allophycocyanin (APC)-conjugated anti-B220 (RA3-6B2), anti-IgM (II/4I), anti-c-Kit (2B8), and anti-CD44 (IM7), PE/Cy7-conjugated anti-Sca1 (D7), anti-CD25 (PC61.5), anti-CD24 (M1/69), and anti-IL-7R (A7R34), Biotin-conjugated anti-CD4 (RM4.5), anti-CD8 (53-617), anti-Ter119 (Ter119), and anti-BP-1 (6C3), PERCp-Cyanine5.5-conjugated anti-CD45.2 (104), and eFluor450-conjugated anti-Ly-6D antibodies were from eBioscience. PE-conjugated anti-CD43 (S7), anti-TCRγδ (GL3), anti-CD8 (53-6.7), and anti-NK1.1 (PK136), Biotin-conjugated anti-B220 (RA3-6B2), anti-CD11b (M1/70), and anti-Gr.1 (RB6-8C5) antibodies, and APC/Cy7-conjugated strepta-vidin were from BD Biosciences. APC/Cy7-conjugated anti-CD45.2 (104) and anti-B220 (RA3-6B2) antibodies were from Biolegend. APC-conjugated streptavidin was from Jackson ImmunoResearch. For staining of cell surface molecules, anti-bodies were used at dilution of 1:400. Samples were applied to a flow cytometer (LSR II, Becton Dickinson) for FACS analysis or cell sorter (FACSAira IIIu, Becton Dickinson) for sorting. Data were collected and analyzed using FACSDiva software (Becton Dickinson) or FlowJo software (Tree Star). FACS gating strategies are presented in Supplementary Fig. 17.

**Intracellular staining of pSTAT5.** Single-cell suspension from the BM of the experimental and control mice were prepared, followed by RBC lysis, and fixed with 2% PFA (Electron Microscopy Sciences Cat15710) at 37 °C for 10 min. Cells were washed twice with PBS and then slowly permeabilized with ice-cold 95% MeOH while vortexing on ice for 20 min. Cells were washed twice with PBS and incubated with Fc-block for 15 min on ice and then stained with PE/Cy7-con-jugated anti-B220, PE-conjugated anti-CD43, and AlexaFluor 647-conjugated anti-phospho-STAT5 (pY694, BD Biosciences), or mouse IgG₁,κ isotype control (BD Biosciences). PE/Cy7-conjugated anti-B220 and PE-conjugated anti-CD43 were used at dilution of 1:400. AlexaFluor 647-conjugated anti-phospho-STAT5 was used at dilution of 1:100.

**BM-derived progenitor differentiation on OP9 stromal cells.** BM Lin⁻ (B220⁻Mac1⁻Gr.1⁻CD4⁻CD8⁻Ter119⁻) progenitors were co-cultured with OP9 cells as previously described with minor modifications[69]. Briefly, BM cells from the experimental and control mice were treated with Gey's solution to lyse RBCs and resuspended in PBS with 2% FBS. The cells were stained with a combination of biotin-conjugated antibody cocktail including anti-B220, anti-CD4, anti-CD8, anti-CD11b, anti-Gr-1, and anti-Ter119. Cells were then incubated with streptavidin-conjugated microbeads (BD Biosciences 557812) and passed through LS magnetic columns (Miltenyl Biotech 130-042-401). The flow-through containing the Lin⁻ BM cells were seeded in 12-well plates at 2–5 × 10⁴ cells per well with ~85% confluent of OP9 cells and with IL-7 (10 ng/ml), Flt-3L (5 ng/ml), and SCF (5 ng/ml). Non-trypsin passages were performed at 4 to 5-day intervals. On day 9, suspension cells were collected, filtered through a 40-μm cell strainer and stained with fluorescence-conjugated anti-B220 and anti-CD11b (Mac-1). Total live cell numbers were determined by trypan blue staining.

Differentiation of lineage-negative BM cells on OP9 cells following acute deletion of PLCγs: Lin⁻ BM cells from the CD45.1⁺ recipients of CD45.2⁺ BM with the indicated genotype were co-cultured with OP9 cells in the presence of IL-7 (10 ng/ml) and Flt-3L (5 ng/ml) for 5 days. Then, the cells were further cultured on OP9 in the presence of IL-7 without or with 4-OHT for 4 more days. The cells were collected, filtered through a 40-μm cell strainer and stained with fluorescence-conjugated anti-CD45.2, anti-B220 and anti-CD11b (Mac-1).

**High-throughput BCR heavy chain sequencing.** Total RNA was isolated from the sorted B cells (5 × 10⁴) using RNeasy Plus Mini Kit (Qiagen Cat# 74134) and cDNA of each sample was generated from 50 ng RNA by using Sensiscript RT kit (Qiagen Cat# 205211). A two-round PCR strategy was used to amply IgM H chains from B cell samples. Briefly, in the 1st round PCR, a degenerate primer mixture 5′ Trp-VHcon1-4 (5′ Trp-VHcon1: CCTCTCTATGGGCAGTCGGTGATGAGGTG CAGCTGCAGGAGTCTGG, 5′ Trp-VHcon2: CCTCTCTATGGGCAGTCGGTG ATGAGGTGCAGCTGCAGGAGTCAGG, 5′ Trp-VHcon3: CCTCTCTATGGGC AGTCGGTGATGAGGTGCAGCTGCAGCAGTCTGG, 5′ Trp-VHcon4: CCTCTCTATGGGCAGTCGGTGATGAGGTGCAGCTGCAGCAGTCAGG)[70] and 3′ Linker-mCμ (3′ Linker-mCu: TGCGAAGTCGACGCTGAGGAGACGAA GACATTT GGGAAGGACTGAC)[71] were used to amplify mouse B cell IgM H chain. For second round PCR, the same 5′ primers and a 3′A-barcode-linker primer with different barcode sequence were used for individual samples. The DNA samples were purified using XP Ampure bead purification and then size selected using the Pippin Prep (Sage Sciences). The samples were run on a 2% DF marker L Pippin Prep cassette with the size range set to elute from 300 to 600 bp. The concentrations of samples were determined by DNA High Sensitivity chip (Agilent Bioanalyzer) and equal amounts of DNA from each sample were pooled. The libraries (30 pg pooled DNA) underwent an emulsion PCR on Ion Torrent OneTouch2, prior to enrichment. The enriched libraries were loaded onto an Ion Torrent 314v2 chip for sequencing. Barcoded sequences were separated using Geneious software or custom analysis scripts. Trimmed sequences were uploaded onto IMGT/HighVQuest server for alignment to germline IgH VDJ regions.

**BrdU incorporation assay.** Pro-B cells (B220^lo CD43⁺CD24⁺BP-1⁻) were sorted from the experimental and control mice and cultured in a 24-well plate with OP9 cells in the presence of IL-7 (10 ng/ml). On day 5, pro-B cells were pulsed with BrdU (2 μM, BD Pharmingen) for 2 h, fixed and stained with anti-BrdU antibodies. BrdU incorporation was determined using the APC-BrdU flow kit according to the manufacturer's instruction (BD Pharmingen).

**Proliferation and cell-cycle analysis.** BM cells from Rag1⁻/⁻ mice were cultured in vitro with IL-7 (10 ng/ml) for 7 days. The IL-7 culture-derived pro-B cells were washed and treated with the indicated inhibitors or DMSO in the presence of IL-7 for 48 h. Cells were harvested and followed by propidium iodide (Sigma) staining for cell-cycle analysis or pulsed for 16 h with ³H-thymidine (1 μCi per well). Thymidine-pulsed samples were collected with the use of a MACH III harvester (TOMEC, Hamden, CT) and the incorporation of ³H-thymidine was determined with a Wallac MicroBeta TriLux scintillation system (PerkinElmer, Waltham, MA).

**TUNEL assay.** BM cells from Rag1⁻/⁻ mice were cultured in vitro with IL-7 (10 ng/ml) for 7 days. The IL-7 culture-derived pro-B cells were washed and treated with the indicated inhibitors at different concentrations or DMSO in the presence of IL-7 for 18 h. Cells were then fixed, permeabilized, and labeled with fluorescein-conjugated dUTP according to the manufacturer's instructions (In situ Cell Death Detection Kit, 11-684-795-910; Roche). Percentage of TUNEL positivity was determined by FACS analysis.

**Metabolism assays.** BM cells from Rag1⁻/⁻ mice were cultured in vitro with IL-7 (10 ng/ml) for 7 days. The IL-7 culture-derived pro-B cells were washed and treated with the indicated inhibitors or DMSO in the presence of IL-7 for 18 h. Cells were then harvested and resuspended in non-buffered media with IL-7 (10 ng/ml). OCR and ECAR were measured under basal conditions and in response to oligomycin A (1 μg/ml) (Sigma), FCCP (3 μM) (Sigma), and antimycin (10 μM) (Sigma) using Seahorse Bioscience Extracellular Flux Analyzer (Seahorse Bioscience).

**Immunoprecipitation and western blotting analysis.** BM cells from Rag1⁻/⁻ mice were cultured with IL-7 (10 ng/ml) for 7 days. The IL-7 culture-derived pro-B cells (10 × 10⁶) were washed, starved in RPMI-1640 medium with 0.1% BSA for 6 h, and stimulated with IL-7 (100 ng/ml) for the indicated times. Cell lysates were immunoprecipitated with anti-PLCγ2 antibodies (Q-20, sc-407, Santa Cruz Biotechnology) and subjected to western blot analysis with anti-phosphotyrosine antibodies (4G10, Millipore). For inhibition experiments, cells (2 × 10⁶ per ml) were pre-treated with the indicated inhibitors or DMSO at 37 °C for 10 min and then stimulated with IL-7 (100 ng/ml) for 90 min. For PdBu and ionomycin sti-mulation, cells (1 × 10⁶ per ml) were starved in RPMI-1640 medium with 0.1% BSA for 6 h and then stimulated with PdBu, ionomycin, or PdBu plus ionomycin for the indicated times. Cell lysates were subjected to western blot analysis with the indicated antibodies. Anti-phospho-S6 (Ser235/236, D57.2.2E), anti-phospho-S6 kinase (T389, 108D2), anti-phospho-mTOR (S2448), anti-phospho-AKT (S473, D9E), anti-phospho-Akt (T308, 9275), and anti-phospho-PLCγ1 (Y783) antibodies were purchased from Cell Signaling Technology. Anti-phospho-ERK (E4) and anti-Akt (H-136) antibodies were from Santa Cruz Biotechnology. Anti-Actin anti-bodies were from Millipore. For western blot analysis, anti-actin antibodies were used at dilution of 1:1000, other primary antibodies were used at dilution of 1:1000, and secondary antibodies were used at dilution of 1:5000. Western blot band intensities were quantified using ImageJ software and the highest tense band of each row was set as 1.0. Full length uncropped western blots are presented in Supplementary Fig. 18.

**RNA-seq.** CLPs (Lin⁻IL-7R⁺) and B cell progenitors (B220⁺IgM⁻IL-7R⁺) were sorted from three experimental and control mice. Total RNA was prepared using TRIzol LS (LifeTechnologies). Overall, 10–50 ng of total RNA was used to purify mRNA by NEBNext poly(A) mRNA Magnetic Isolate Module (New England Biolabs), and then converted into libraries using NEBNext Ultra RNA library Prep kit for Illumina (New England Biolabs). Libraries were quantified by Qubit-Fluorometer (ThermoFisher) and Kapa Library Quantification Kit (KapaBiosys-tems). Average size of the libraries was estimated by D1000 ScreenTape system (Agilent) with region selection from 150 to 900 bp. Equal moles of libraries were mixed and a total library of 1.7 pmoles was sequenced on Illumina NextSeq 500 with NextSeq 500/550 v2 kit.

**RNA-seq data analysis.** Raw reads of RNA-seq were preprocessed on the Illumina BaseSpace sequence hub and aligned to the mouse reference genome Mus mus-culus/mm10 (RefSeq) using the aligner STAR. Differentially expressed transcripts with FDR ≤0.05 were identified using the DESeq2 program[72] as implemented on the Illumina BaseSpace hub. Functional enrichment and pathway analysis of gene lists was performed using the Ingenuity Pathways Analysis software (Qiagen) and the DAVID Bioinformatics Resources (https://david.ncifcrf.gov/).

**Intracellular Ca²⁺ measuring.** B cell progenitors were derived from Rag1-deficient BM by culturing in vitro with IL-7 for about 7 days. The B cell progenitors were

washed and starved in RPMI-1640 without sodium bicarbonate (Sigma-Aldrich; R7388) and IL-7 for 3 h prior to loading with 1 μM Oregon Green 488 BAPTA-2 (ThermoFisher; O6809) in the same medium for 1 h at 37 °C. The cells were washed and resuspended in medium without or with 25 ng/ml of IL-7, and then were plated onto μ-Slides (Ibidi USA; 80426) coated with 5 μg/ml Cell Tak (Corning; 354240). Relative intracellular calcium levels were evaluated by fluorescence lifetime microscopy (FLIM) using an inverted AxioObserverZ.1 microscope (Zeiss) equipped with a LuxX Diode 488 nm laser (Omicron-Laserage), intensifier (Lambert Instruments), CSU-X spinning disk (Yokogawa), coolsnap EZ CCD camera (Photometrics), and a 1.3 NA ×40 oil objective. Images were acquired and analyzed using Slidebook software (Intelligent Imaging Innovation).

**Statistical analysis**. All statistical analysis was performed with the two-tailed unpaired student $t$ test.

**Data availability**. Sequence data that support the findings of this study have been deposited in GEO with the primary accession code GSE89352. However, BCR heavy chain sequence data are not deposited and are available from the corresponding author upon reasonable request.

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

## Acknowledgements
This work is supported in part by NIH Grants AI105887 (H.C.), AI101407 (H.C.), CA176624 (H.C.), NS064599 (H.C.), AI079087 (D.W.), and HL130724 (D.W.).

## Author contributions
M.Y. contributed in research design, performed the research, analyzed the results, and wrote a part of the manuscript. Y.C., H.Z., Y.Z., G.F., and W.Z. performed some experiments. U.B. reviewed the manuscript and helped the high-throughput BCR heavy chain sequencing experiment. P.A. and A.T. helped the high-throughput BCR heavy chain-sequencing experiment. G.N. helped RNA-seq data analysis. C.G. helped the Ca²⁺ measuring experiments. N.Z. helped the RNA-seq experiments and RNA-seq data analysis. H.C. helped RNA-seq data analysis and critically reviewed the manuscript. R.W. provided intellectual input, supervised the study, and critically reviewed the manuscript. D.W. conceived and supervised the study, analyzed the results, and wrote the manuscript.

## Additional information

**Competing interests:** The authors declare no competing financial interests.

