## [Peer review file · Nature Communications]

Reviewers' comments:

Reviewer #1 (Remarks to the Author):

In this manuscript, Yu et al. demonstrate that double PLCg1-PLCg2 deficiency arrests B cell development at the pre-pro-B stage by blocking IL-7-induced activation of mTOR. This is a well-written, novel and elegantly presented manuscript that provides a wealth of compelling data convincingly supporting the main conclusions. The following specific comments are provided to enhance this study.

Specific comments

1. Kalaitzidis et al. (Cell Stem Cell. 2012; 11:429-439) demonstrate that Raptor deficiency results leads to an increase of B220+IgM-CD43+ pro-B cells in the bone marrow. The authors should discuss these data in relationship to their findings and try to reconcile discrepancies.
2. The role of PLCg1 and PLCg2 in B cell lymphopoiesis was evaluated by using an IFN-inducible MxCre mouse model. Does IFN-inducing poly(I:C) treatment influence B cell lymphopoiesis? In Figure 2a, it would be important to clarify whether Rag1^{-/-} animals were treated with poly(I:C) as PLCg⁻-DKO mice were. Along the same lines, it would be important to clarify whether MxCre induction is reversible or irreversible. If reversible, how long does the effect persist? This is relevant to in vitro experiments shown in Figure 9d.
3. The elevated variability of data shown in Figure 2a,b could result from the small sample size. It is advisable to increase the number of animals.
4. In Supplementary Figure 4, the authors show that PLCg deficiency blocks IL-7-dependent B cell differentiation by using a tamoxifen-inducible ERCre system. Does the observed reduction of B cells result from treatment with 4-OHT? Given the significant reduction of B cells prior to 4-OHT treatment in day-9 cultures (5+4), it would be reassuring to show data obtained at the same time point but in the absence of 4-OHT.
5. In Figure 2c, MFI values should be provided together with specific values for pre-pro-B cells to unequivocally demonstrate that surface IL-7R levels are similar in different genotypes.
6. In Figure 2g, examples of single-KO should be shown. For each experiment, statistics and number of animals used should be indicated.
7. In Figure 4d-g, heat maps nicely show genes differentially expressed by PLCg-DKO progenitor B cells. A bar plot or heat map summarizing top differentially expressed pathways/biological processes would provide a more comprehensive view of the changes occurring in the absence of PLCg signaling.
8. In Figures 4d and 4g, PLCg-DKO animals show reduction of mTOR-dependent gene expression. This finding should be further discussed and validated by qRT-PCR. Expression of mTOR-related molecules central to mTORC1/mTORC2 function (e.g. Raptor, Rictor) should be shown.
9. In Figure 6a,b, it would be important to clarify whether there is any difference in pS6 expression levels between untreated B cell progenitors from different KO animals.
10. On page 21, line 464, AKT inhibition by MK-2206 is commented as being ineffective to block IL-7-induced S6 phosphorylation in pro-B cells (Figure 8b). However, this was only true when low MK-2206 concentrations were used. Was this choice dictated by induction of cell death? If possible, a titration of MK-2206 should be shown. In addition, band intensities should be quantified and time of treatment added to the figure legend.

Minor comments

1. Figure legend corresponding to Suppl. Figure 1: "Sac1-" should be "Sca-1"
2. Figure 1b. "a-actin" should be "alpha-actin".
3. Figure 1h. Addition of fraction names in the first dot plot would help readers in the interpretation of the data.
4. Figure 2f. Number of animals and cells needs to be provided. Most of the main figures lack this information.
5. Error bars can be hardly seen in most of the figures.
6. Figure 3d,e. To better discriminate experimental conditions, authors could label "IL-7-expanded" and "non-expanded" B cells.
7. Page 11, line 228-229. "we study..." please change to past tense.
8. Page 13, line 286. Please omit "et al." from "EBF1, Tcf12, Notch1, et al...". Moreover, gene names should be italicized.
9. Figure 4b. A gene name is missing after CD72.
10. Figure 4a and 4c. Font size for X/Y axes values should be increased.
11. Page 14, line 314. mTOR is currently defined as mechanistic target of rapamycin rather than "mammalian target of rapamycin".
12. Figure 6d. Please add MFI values to histogram plots.
13. Figure 6, 7 and 8. Authors should provide more information as to how band intensities were calculated and which population was used as reference.
14. Supplementary figures 6a,b are quoted in the text before supplementary figure 5c.
15. The Discussion is perhaps too long and does not include the possible clinical implications of the main findings.

Reviewer #2 (Remarks to the Author):

Yu et al. study B-cell development from hematopoietic stem cells (HSC) in PLCg1-deficient, PLCg2-deficient and PLCg1/PLCg2-double deficient mice. They find that early B-cell development is inhibited in double deficient hematopoietic cells first at the stage of pre-proB-cells, rendering these cells unresponsive to the cytokine IL7, for which they continue to express the receptor, IL7a. The double deficiency also leads to a block in T-cell development at the DN stage. The authors show these blocks of development by bone marrow-, i.e. Double PLCg-deficiency-affected HSC-transplantations into irradiated recipients, with cells from appropriate control mice. It would be good to see these analyses also being done in the original PLCg1/PLCg2-double-deficient, cre-activated, floxed mice. The authors also study PLCg-deficient progenitors developing in these transplanted mice by in vitro cultures of lin- progenitors, which are known to develop more mature B-lineage cells.

It would also have been useful to use a more complete set of markers to define the different stages of hematopoietic and B-lymphopoietic cell development, i.e. to lin⁻ CD48+sca1+c-kit⁺ MPP, lin⁻ c-kit+IL7Ra+flt2+CD27⁺ CLP, B220+c-kit+flt2+CD24-BP-1-CD43+CD19⁻ pre-proB-cells, c-kit+flt2+CD24+BP-1-CD19+CD25⁻ early proB-cells and c-kit+flt2-CD24+BP-1+CD19⁺ proB-cells (see Osmond et al. Immunol. Today 1998, 19:65-8), c-kit-CD19+CD25+IgM⁻ preB (also called preBII-cells) and CD25-CD93+IgM⁺ immature B-cells, as well as a more specific analysis of the myeloid cell compartments, using at least not only Mac-1, but also c-fms, Gr-1, CD11c and F4/80. IL7-deficiency arrests at the CLP to CD19-pre-proB-cells stages, but RAG-deficient progenitors arrest at the CD19⁺ pro(preBI)-like stage. Therefore, the experiments presented in the paper probe TWO stages of B-cell development, visible in transplantation experiments at the earlier pre-proB-cell stage, and in OP9/IL7 culture experiments and inhibitor experiments with RAG-deficient cells at the later pro(also called preBI) cell stage. It extends their claim of IL7-PLCg-DAG-mTOR signaling to two stages of B-cell development. This must be properly discussed. Also, why did the authors not do the experiments (from line 342 onwards) also with pro(preBI)cells from wildtype mice, since these cells can be cultured on stromal cells and IL7 for extended periods of time? This would have allowed to distinguish the roles of IL7 and of kit-ligand (OP9) signaling in proliferation and survival of the pro(preBI) cells and the PLCg1/PLCg2 double, as well as the PLCg2 single deficiency on proliferation, survival and differentiation of the pro(preBI)-cells.

Lines 169-196: In order to further define the defect, PLCg1/PLCg2-double deficient lin⁻ bone marrow progenitors were cultured on OP9 stromal cells and IL7. While non-double deficient control cells developed, as expected, to B-lineage cells (hopefully CD19⁺), and to few myeloid cells, the double-deficient progenitors failed to develop B-lineage cells, but myeloid cells developed. In numbers, how many myeloid cells, and of which phenotype (see above)? Do the authors think that IL7 (or the M-CSF-deficient OP9 stromal cells) induced this myeloid cell differentiation? Did the double deficiency increase myeloid development, and why did myeloid cells develop, although the cultures did not contain myeloid differentiation-inducing cytokines – notably M-CSF, which is not made by OP9 stromal cells? M-CSF-producing stromal cells (e.g. ST2) and addition of cytokines, such as M-CSF, GM-CSF, G-CSF and others should be tested, and the response quantitated by quantitation of developing phenotypically-defined myeloid cells.

Similar questions arise from the results of the acute in vitro deletion of PLCs and their ensuing cell productions on OP9 and IL7. Again, the non-deficient controls, developing the expected B-lineage cells, developed some myeloid cells, although it would, again, be good to see more myeloid markers. Again, the double PLCg deficiency, blocking IL7-stimulated B-cell development, appears to have allowed (increased??) myeloid (which) development? Again, as suggested above, myeloid development-inducing culture conditions should be used to test a possible (new) preponderance of the double deficient cells for myeloid development.

Although the authors show that the first defect in B-cell development occurs at the pre-proB-cell stage and involves signaling of IL7 via the IL7Ra, they find, that the small number of more differentiated early and late proB-cells (also called preBI-cells), which still develop from the already defective pre-proB-cells in vivo are also defective: they do not proliferate and do not establish preB-cell lines, as PLCg-non-defective proB- (also called preBI-cells) do. Why? Is this an indication for the PLCg-dependent IL7 signaling at a later developmental stage of B-cells? This should be discussed.

Lines 227-239: Since the pre-proB-cells of double PLCg-deficient mice are defective in development to more mature stages, it appears that the few pro(preBI)-cells, which still are formed, retain the very same defect of IL7 signaling.. The authors try to investigate, whether the decrease in numbers of more differentiated B-lineage cells could be the result of a defect of the ability to proliferate and/or to survive. IL7 has been seen to induce increased survival of pro(preBI)-cells, while the additional action of stromal cells, and kit-ligand provided by them, stimulates their proliferation. The experiments lack important controls: cells with either only IL7, or only stromal cells. It appears, therefore, premature to expect the PLCg-deficiency, and the actions of the PLCg-inhibitors to be merely, or at all, affecting proliferation. Also, myeloid

differentiation results need quantification and specifications of states of differentiation.

Lines 240-271: The same critique holds for the experiments, where wild-type lin- BM progenitors are tested with PLCg-inhibitors and Jak3-inhibitors in these in vitro cultures. Furthermore, a more detailed marker analysis of the IL7R+ cells (lines 264 ff), which are expected to be heterogeneous and may not all react the same way, and the cells after in vitro treatments is needed to evaluate these experiments.

Lines 313-354: It is trendy, but still comes as a surprise, that the authors studied IL7-dependent metabolism. The critique of incomplete characterization of the studied IL7R+ cells, and the in vitro conditions (IL7 only, IL7+OP9, OP9 only) also applies here. Thus, the conclusions drawn from these experiments on lines 339 and 340 are at least premature.

As stated above, the RAG-deficient cells studied here are CD19+pro(preBI)-cells. The data on phosphorylation of PLCg1, PLCg2, ribosomal protein S6 and mTOR are central results to the claims of the authors, that IL7 uses this pathway, here in pro(preBI)-cells. Again, OP9+IL7 and OP9 alone are missing conditions.

The authors should mention, at least at this point of their studies, that the small number of PLCg-deficient pre-proB-cells remaining in the transplanted hosts for biochemical studies of the type they present, are inferior to the results with RAG-deficient pro(preBI)-cells. Again, OP9+IL7 and OP9 alone are missing conditions.

Lines 384-end of results section: here, RAG-deficient pro(preBI)-cells are the target of IL7-induced responses – unfortunately again without OP9 stromal cells alone or together with IL7 - in well-conducted, convincing studies. The reviewer admits lack of expertise to judge e.g. the target specificity of these inhibitors. The reviewer is aware of studies, in which genetic mutations in the signaling pathways are used to imply a particular molecular target as operative – but it is unreasonable to suggest such mutational analyses of the PLCg1/PLCg2-DAG-mTOR-S6 pathway for the present work.

The submitted work has two messages:

- 1) PLCg1/PLCg2-double deficiency of hematopoietic progenitors severely impairs IL7-dependent B-cell development at the transition from CLP to pro-preB-cells, while allowing (maybe STAT5-dependent, ebf1-pax5-mediated) differentiation.
- 2) PLCg1/PLCg2-double deficiency of pro(preBI)-cells, either developed from double deficient pre-proB-cells, or induced by PLCg-inhibitors in pro(preBI)-cells use a novel signaling pathway from the IL7R via PLCg1/PLCg2-DAG-mTOR-S6.

The authors should consider: Since the submitted paper is very long, why not make two papers out of one, including, what appear reasonable controls.

I think that the STAT5-independent, PLCg-dependent action of IL7 is a novel discovery, important for a better understanding of the role of IL7 in B-lymphoid development. Provided, that the authors deal with the points (which are made to improve the paper) I recommend publication.

Reviewer #3 (Remarks to the Author):

This manuscript by Yu et al explores the unique and redundant contribution of PLCg1 and 2 in early B cell development by using single and dual-deficient mice. A very thorough analysis of B cells in these mice leads to the novel and interesting discovery that PLCg1/2 enzymes are essential contributors to IL-7R signaling and, therefore, to B cell responses to IL-7. This contribution is highlighted by finding that mice with PLCg1/2 double deficient hematopoietic cells are largely unable to generate pro-B and pre-B cells and are consequently deprived of immature and mature B cells. Extensive biochemical characterization, moreover, uncovers that under IL-7R signaling, PLCg1/2 activates mTOR by a previously unrecognized PKC-dependent and Akt/Tsc/Rheb-

independent pathway.

These findings provide new insights into the mechanism of B cell development and IL-7R signaling, insights that can also help with understanding the root of some B cell deficiencies and B cell leukemia. It is of interest noting that past studies of mice with deficiency in one or the other PLCg forms have indicated only a small contribution of these phospholipid-specific enzymes to early B cell development and none to IL-7R signaling. In fact, these partial deletions have resulted at time in increased B cell responses to IL-7 because they led to the accumulation of B cell precursors that heavily depend on this cytokine. The perseverance of this group in pursuing the function of PLCg1/2 in B cell development has ultimately paid off with the present study, which leads to important novel discoveries.

The study is extensive and thorough, investigating PLCg1/2 at the cellular, biochemical, metabolic, and genetic levels. The data indicating the PLCg1/2 enzymes operate under IL-7R signaling and that their deficiency recapitulates IL-7R defects are convincing. The use of complementing approaches, such as performing both ex-vivo and in-vitro culture studies and analyzing B cells with either genetic ablation or pharmacological inhibition of PLCg1/2, provide compelling evidence to support the authors' conclusions. Statistical analyses are properly provided and for experiments with more variability (e.g., western blots), the experiments were repeated at least 3 times and representative data reported.

However, the (heavy and redundant) writing style of the paper detracts from the importance of the discovery and the paper could be strengthened by streamlining the description of the results.

Additional suggestions for strengthening conclusions are as follow:

1) The authors use high-throughput sequencing to determine the usage frequency of VHJ558 family genes relative to VH7183 family genes (Fig. 2f). The use of these data to support the conclusion that PLCg1/2 double deficient pro-B cells have defective IL-7 responses is appropriate. However, readers and B cell biologists in particular may be disappointed by the scarce use of these data, and might be interested in knowing whether PLCg1/2-deficient pro-B cells display a general VDJ recombination defect or not. This knowledge provides a clearer overall picture of the contribution of the PLCg pathway in early B cell development. Was the VH7183 usage in PLCg-deficient pro-B cells normal or decreased relative to PLCg-sufficient pro-B cells? Is this similar to that of Jak3^{-/-} pro-B cells?

2) Conclusions from experiments in which B cell progenitors were treated in cultures with PLCg and PKC inhibitors (Figs. 3 and 5) would be greatly strengthened by using control inhibitor(s) for a pathway irrelevant in these cells. Sometimes inhibitors have unspecific effects on cell survival, proliferation and/or metabolism and showing that an inhibitor for an unrelated pathway does not affect these processes will strengthen the conclusions of experiments that used inhibitors for the pathway of interest. There is also an issue with specificity of the inhibitors: would that be possible to show that U73122, Edelfosine and Manolidol do indeed inhibit PLCg more directly? Do they at least inhibit Ca⁺⁺ release in these pro-B cells?

3) The biochemical analyses are exceptionally well done, especially when considering the limited amount of pro-B cells available for these studies. However, analyses of PLCg1/2 double deficient pro-B cells are too limited. It is understandable that the amount of these cells is prohibitive for western blots, but it is unclear why the investigators did not take advantage of phosphoflow studies. Phosphoflow analyses of pS6 are quite robust and it would be important to show pS6 levels in PLCg1/2 double deficient pro-B cells treated or not with IL-7. These experiments should be feasible.

4) The results of experiments showed in Fig. 7b,h in which Rag1^{-/-} pro-B cells treated with the PKC agonist PDBu augment S6 and mTOR activation and cell proliferation complement greatly the inhibitor studies. Following this line of thoughts and considering PKC is downstream of PLCg in the pathway uncovered in this study, it should be possible to treat PLCg1/2 double deficient pro-B cells

with PDBu (+/- ionomycin) and find increase of pS6 (by flow for instance), and cell survival and proliferation independent of IL-7.

Reviewer#1:

Reviewer #1 thinks that our paper “is well-written, novel and elegantly presented and provides a wealth of compelling data convincingly supporting the main conclusions.” The reviewer also gives several specific suggestions and insightful comments. We have revised our manuscript based on these important suggestions.

Response to specific comments:

1 Kalaitzidis et al. (Cell Stem Cell. 2012; 11:429-439) demonstrate that Raptor deficiency results leads to an increase of B220⁺IgM⁻CD43⁺ pro-B cells in the bone marrow. The authors should discuss these data in relationship to their findings and try to reconcile discrepancies.

We thank the reviewer for pointing out the previous publication. This report states that Raptor deficiency leads to an increase of B220⁺IgM⁻CD43⁺ pro-B cells in the BM. However, this publication does not present data showing an increase in the absolute number or percentage of pro-B cells. We suspect that due to marked reduction of pre-, immature and mature B cells in the BM, the percentage of pro-B cells is increased in this published study. Consistent with our current study, a recent article demonstrates that B cell-specific deletion of Raptor blocks early B cell development at the pre-pro-B cell stage, resulting in the reduction of the number of early pro-B (fraction B) (J Immunol 2016, 197:2250). Now, we have discussed the newly published findings in the Discussion section (page 28) of the revised manuscript.

2 The role of PLCg1 and PLCg2 in B cell lymphopoiesis was evaluated by using an IFN-inducible MxCre mouse model. Does IFN-inducing poly(I-C) treatment influence B cell lymphopoiesis? In Figure 2a, it would be important to clarify whether Rag1^{-/-} animals were treated with poly(I:C) as PLCγ-DKO mice were. Along the same lines, it would be important to clarify whether MxCre induction is reversible or irreversible. If reversible, how long does the effect persist? This is relevant to in vitro experiments shown in Figure 9d.

Mx1Cre mice have the Cre recombinase gene under control of the IFN-inducible Mx1 promoter and are commonly used to study the effect of inducible deletion of a specific gene. Although deletion of a specific gene by poly(I-C) injection is irreversible, the induction of IFN and its effect on hematopoiesis, including lymphopoiesis, are reversible. As shown in the original publication of the Mx1-Cre system, the populations of B and T lymphocytes appear

normal again following poly(I-C) injection (Science 1995, 269: 1427-1429). A recent study has demonstrated that poly(I-C) injection has a transient effect on HSCs but HSCs recover 8 days after the injection (Stem Cell Reports 2016, 7: 11–18). Our studies also demonstrate that the effect of poly(I-C)-induced interferon is transient (data not shown). Therefore, to avoid the potential side effect caused by poly(I-C) injection and subsequent IFN production, experimental and control mice were analyzed 3 weeks after the last poly(I-C) injection. In **Fig 2a**, BM from poly(I-C)-treated experimental and control mice were transplanted into lethally irradiated congenic wild-type CD45.1⁺ mice. Six to eight weeks after transplantation, Lin⁻ BM cells from the recipients that were not treated with poly(I-C) were used for the experiments. Thus, any possible effect of poly(I-C) administration and subsequent IFN production was excluded. Poly(I-C)-induced deletion of a specific gene is irreversible whereas the induction of IFN and its effect on hematopoiesis are transient and reversible. The information that donor mice were sacrificed 3 weeks after the last injection of poly(I-C) for BM isolation has been emphasized in the Methods section (page 32) of the revised manuscript.

3. The elevated variability of data shown in Figure 2a, b could result from the small sample size. It is advisable to increase the number of animals.

We thank the reviewer for this advice. The results in **Figs 2a** and **2b** were obtained from 6 independent experiments.

4. In Supplementary Figure 4, the authors show that PLCg deficiency blocks IL-7-dependent B cell differentiation by using a tamoxifen-inducible ERCre system. Does the observed reduction of B cells result from treatment with 4-OHT? Given the significant reduction of B cells prior to 4-OHT treatment in day-9 cultures (5+4), it would be reassuring to show data obtained at the same time point but in the absence of 4-OHT.

For **Supplementary Fig 5** of the revised manuscript, we indeed performed the experiments at the same time point in the absence (middle panels) and presence (lower panels) of 4-OHT. Without 4-OHT treatment, PLCγ1^{fl/fl}PLCγ2^{-/-} progenitors produced a reduced number of B cells compared to the controls (middle panels). However, following 4 days of 4-OHT treatment, PLCγ1^{fl/fl}PLCγ2^{-/-} progenitors completely failed to produce B cells (lower panel). Thus, acute deletion of PLCγ genes completely blocks IL-7-dependent B cell production in vitro.

5. In Figure 2c, MFI values should be provided together with specific values for pre-pro-B cells to unequivocally demonstrate that surface IL-7R levels are similar in different genotypes.

We thank the reviewer for this suggestion. MFI values have been included in **Fig 2c** of the revised manuscript.

6. In Figure 2g, examples of single-KO should be shown. For each experiment, statistics and number of animals used should be indicated.

As suggested, the data of IL-7-induced BrdU incorporation in PLCγ1^{-/-}PLCγ2^{+/+} or PLCγ1^{+/+}PLCγ2^{-/-} B progenitors has been added to **Fig 2g** and described in the Results section (page 11) of the revised manuscript. We apologize for missing statistics and numbers of mice used

for some experiments. Now, statistics and numbers of mice used for each experiment have been indicated in the Figure Legends of the revised manuscript.

7. In Figure 4d-g, heat maps nicely show genes differentially expressed by PLCg-DKO progenitor B cells. A bar plot or heat map summarizing top differentially expressed pathways/biological processes would provide a more comprehensive view of the changes occurring in the absence of PLCg signaling.

We thank the reviewer for the suggestion. A bar plot summarizing top activated and inhibited pathways in PLC γ 1^{-/-}PLC γ 2^{-/-} pro-B cells has been added to **Supplementary Fig 7a** and described in the Results section (page 15) of the revised manuscript.

8. In Figures 4d and 4g, PLCg-DKO animals show reduction of mTOR-dependent gene expression. This finding should be further discussed and validated by qRT-PCR. Expression of mTOR-related molecules central to mTORC1/mTORC2 function (e.g. Raptor, Rictor) should be shown.

In **Figs 4d** and **4g**, ingenuity pathway analysis demonstrated that B cell lineage-affiliated and IL-7 signaling pathway-regulated genes were significantly downregulated in PLC γ 1^{-/-}PLC γ 2^{-/-}, relative to PLC γ 1^{+/+}PLC γ 2^{+/+}, B cell progenitors. The reduction of some of the top listed genes was confirmed by qRT-PCR. However, the expression of Raptor and Rictor was not significantly changed in PLC γ 1^{-/-}PLC γ 2^{-/-} relative to control B cell progenitors. The results have been included in **Supplementary Fig 7b** and described in the Results section (page 15) of the revised manuscript.

9. In Figure 6a,b, it would be important to clarify whether there is any difference in pS6 expression levels between untreated B cell progenitors from different KO animals.

In the revised manuscript, we state that “the basal level of S6 phosphorylation was markedly decreased in PLC γ 1^{-/-}PLC γ 2^{-/-} relative to the corresponding PLC γ 1^{+/+}PLC γ 2^{+/+}, PLC γ 1^{-/-}PLC γ 2^{+/+} or PLC γ 1^{+/+}PLC γ 2^{-/-} B cell progenitors (**Fig. 6c**)” on Page 18.

10. On page 21, line 464, AKT inhibition by MK-2206 is commented as being ineffective to block IL-7-induced S6 phosphorylation in pro-B cells (Figure 8b). However, this was only true when low MK-2206 concentrations were used. Was this choice dictated by induction of cell death? If possible, a titration of MK-2206 should be shown. In addition, band intensities should be quantified and time of treatment added to the figure legend.

As suggested, we indeed examined the effect of three different concentrations of MK-2206 on IL-7-induced S6 phosphorylation. Lower concentrations of MK-2206 had no effect on IL-7-induced S6 phosphorylation (**Fig 8b**), but inhibited BCR-induced S6 phosphorylation (**Supplementary Fig 13**). This information has been emphasized in the Results section (page 22) of the revised manuscript. In addition, the band intensities of **Fig 8b** were quantified and the information regarding the time of treatment was included in the Methods section (page 37) and was also added to the Figure Legend of **Fig 8** of the revised manuscript.

Minor comments

1. Figure legend corresponding to Suppl. Figure 1: “Sac1-“ should be “Sca-1”

The error has been corrected in the revised manuscript.

2. Figure 1b. “a-actin” should be “alpha-actin”.

The error has been corrected in **Fig 1b** of the revised manuscript.

3. Figure 1h. Addition of fraction names in the first dot plot would help readers in the interpretation of the data.

The fraction names have been added to the first dot plot in **Fig 1h** of the revised manuscript.

4. Figure 2f. Number of animals and cells needs to be provided. Most of the main figures lack this information.

The numbers of animals and cells for all experiments are provided in the Methods section. The information requested for **Fig 2f** is in the Method section (page 34) and is also added to the Figure Legend of **Fig 2** of the revised manuscript.

5. Error bars can be hardly seen in most of the figures.

The Figures have been enlarged so that error bars are more obvious.

6. Figure 3d,e. To better discriminate experimental conditions, authors could label “IL-7-expanded” and “non-expanded” B cells.

Figs 3d-g have been labeled as suggested.

7. Page 11, line 228-229. “we study...” please change to past tense.

The error has been corrected.

8. Page 13, line 286. Please omit “et al.” from “EBF1, Tcfl2, Notch1, et al...”. Moreover, gene names should be italicized.

Changes (on page 14 of the revised manuscript) have been made as suggested.

9. Figure 4b. A gene name is missing after CD72.

The gene is PLC γ 2 that has been added to the list in **Fig 4b**.

10. Figure 4a and 4c. Font size for X/Y axes values should be increased.

The font size for X/Y axes values has been increased as suggested.

11. Page 14, line 314. mTOR is currently defined as mechanistic target of rapamycin rather than “mammalian target of rapamycin”.

The error has been corrected.

12. Figure 6d. Please add MFI values to histogram plots.

MFI values have been added to **Fig 6d** of the revised manuscript.

13. Figure 6, 7 and 8. Authors should provide more information as to how band intensities were calculated and which population was used as reference.

The required information is provided in the Methods section (page 37) of the revised manuscript.

14. Supplementary figures 6a,b are quoted in the text before supplementary figure 5c.

The numbering sequence of Supplementary Figures has been adjusted. This issue has been solved.

15. The Discussion is perhaps too long and does not include the possible clinical implications of the main findings.

The discussion is shortened and the possible clinical implications are mentioned.

Reviewer#2:

Reviewer #2 thinks that “the STAT5-independent, PLC γ -dependent action of IL7 is a novel discovery, important for a better understanding of the role of IL7 in B-lymphoid development. Provided we address the points s/he raised, s/he will recommend publication. We have revised our manuscript based on these insightful and important suggestions.

In addition, Reviewer #2 points out that our manuscript is very long and suggests us to split it into two manuscripts. The importance and novelty of our findings are two-fold. If the editor agrees, we are willing to report our findings in two manuscripts. One manuscript describes a critical role of the PLC γ pathway in IL-7-mediated early B cell development. The other manuscript reports that the IL-7 receptor activates mTORC1 through a novel and unconventional PLC γ / DAG/PKC-dependent pathway in controlling early B lymphopoiesis.

Response to specific comments:

1 The authors show these blocks of development by bone marrow-, i.e. Double PLCg-deficiency- unaffected HSC-transplantations into irradiated recipients, with cells from appropriate control mice. It would be good to see these analyses also being done in the original PLCg1/PLCg2-double-deficient, cre-activated, floxed mice.

As suggested, we examined B cell development in MxCrePLC γ ^{fl/fl}PLC γ 2^{-/-} mice three weeks following poly(I-C) injection. The blockade of B cell development at the pre-pro-B stage in the Cre-activated original PLC γ 1/PLC γ 2 double-deficient mice recapitulated the phenotype in the recipients of BM derived from poly(I-C)-treated PLC γ 1^{fl/fl}PLC γ 2^{-/-} mice. These new data were added to **Supplementary Fig 2** and described in the Results section (page 7) of the revised manuscript.

2 The authors also study PLCg-deficient progenitors developing in these transplanted mice by in vitro cultures of lin- progenitors, which are known to develop more mature B-lineage cells. It would also have been useful to use a more complete set of markers to define the different stages of hematopoietic and B-lymphopoietic cell development, i.e. to lin-

CD48+sca1+c-kit+ MPP, lin- c-kit+IL7Ra+flt2+CD27+ CLP, B220+c-kit+flt2+CD24-BP-1-CD43+CD19- pre-proB-cells, c-kit+flt2+CD24+BP-1-CD19+CD25- early proB-cells and c-kit+flt2-CD24+BP-1+CD19+ proB-cells (see Osmond et al. Immunol. Today 1998, 19:65-8), c-kit-CD19+CD25+IgM- preB (also called preBII-cells) and CD25-CD93+IgM+ immature B-cells, as well as a more specific analysis of the myeloid cell compartments, using at least not only Mac-1, but also c-fms, Gr-1, CD11c and F4/80.

As suggested, we determined the numbers of CLPs, BLPs, pre-pro-, early pro- and late pro-, pre-, immature and mature B cells in PLC γ 1/PLC γ 2 double-deficient and control mice. The data were presented in **Figs 1e, 1g and 1i**, and **Supplementary Fig 4**. We also examined the myeloid cell compartments using Mac-1, Gr-1, F4/80, CD11c and other surface markers. The data were added to **Supplementary Fig 1b-e** and described in the Results section (pages 5 and 6) of the revised manuscript.

3 IL7-deficiency arrests at the CLP to CD19-pre-proB-cells stages, but RAG-deficient progenitors arrest at the CD19+ pro(preBI)-like stage. Therefore, the experiments presented in the paper probe TWO stages of B-cell development, visible in transplantation experiments at the earlier pre-proB-cell stage, and in OP9/IL7 culture experiments and inhibitor experiments with RAG-deficient cells at the later pro(also called preBI) cell stage. It extends their claim of IL7-PLC γ -DAG-mTOR signaling to two stages of B-cell development. This must be properly discussed.

IL-7 regulates the expansion of pro-B cells. IL-7 deficiency leads to a severe reduction in the number of pro-B cells, thus arresting early B cell development at the pre-pro-B cell stage. Pro-B cells are highly responsive to IL-7 stimulation, making them particularly suitable for the biochemical study of IL-7 signaling. Rag1 deficiency blocks the expression of the pre-BCR and prohibits pre-BCR-mediated expansion of pre-B cells, arresting B cell development at the pro-B cell stage. Rag1-deficient pro-B cells express IL-7R and are thus entirely suitable for the study of IL-7-mediated signaling and functions. This important point has been discussed (pages 17, 25, and 26) in the revised manuscript.

4 why did the authors not do the experiments (from line 342 onwards) also with pro(preBI)cells from wildtype mice, since these cells can be cultured on stromal cells and IL7 for extended periods of time? This would have allowed to distinguish the roles of IL7 and of kit-ligand (OP9) signaling in proliferation and survival of the pro(preBI) cells and the PLC γ 1/PLC γ 2 double, as well as the PLC γ 2 single deficiency on proliferation, survival and differentiation of the pro(preBI)-cells.

We thank the reviewer for raising this important issue. Signals emanating from the IL-7R and pre-BCR coordinate to regulate the development of pro- and pre-B cells. Our previous study has shown that the PLC γ pathway plays an important role in pre-BCR-mediated functions (EMBO J 2004, 23: 4007). To exclude any potential interference of the pre-BCR with IL-7R signaling, we utilized Rag1-deficient pro-B cells that lack the pre-BCR for biochemically characterizing the role of the PLC γ /mTOR pathway in IL-7R signaling. In addition, both wild-type and Rag1-deficient B cell progenitors respond to c-Kit-ligand (SCF) and can be used for c-Kit signaling. We found c-Kit deficiency did not impair IL-7-mediated production of pro-B cells in vitro (data not shown). The SCF-induced colony formation assay also revealed that PLC γ 1/PLC γ 2 double-deficient BM cells responded normally to SCF. The new data were included in the Results section (page 6 and **Supplementary Fig 1f**) and these important issues were discussed (pages 25 and 26) in the revised manuscript.

5 Lines 169-196: In order to further define the defect, PLCg1/PLCg2-double deficient lin-bone marrow progenitors were cultured on OP9 stromal cells and IL7. While non-double deficient control cells developed, as expected, to B-lineage cells (hopefully CD19+), and to few myeloid cells, the double-deficient progenitors failed to develop B-lineage cells, but myeloid cells developed. In numbers, how many myeloid cells, and of which phenotype (see above)? Do the authors think that IL7 (or the M-CSF-deficient OP9 stromal cells) induced this myeloid cell differentiation? Did the double deficiency increase myeloid development, and why did myeloid cells develop, although the cultures did not contain myeloid differentiation-inducing cytokines – notably M-CSF, which is not made by OP9 stromal cells? M-CSF-producing stromal cells (e.g. ST2) and addition of cytokines, such as M-CSF, GM-CSF, G-CSF and others should be tested, and the response quantitated by quantitation of developing phenotypically-defined myeloid cells.

Lin-BM progenitors cultured on OP9 stromal cells with IL-7 can give rise to a small number of myeloid cells. OP9 stromal cells can produce robust SCF, which contributes to the generation of myeloid cells (PNAS 1999, 96: 9797). The numbers of Mac-1⁺ myeloid cells derived from wild-type, PLCγ single-deficient and PLCγ double-deficient BM were similar and these data are presented in **Fig 2b**. We believe that IL-7 only directs the expansion and differentiation of lymphoid lineage progenitors. Importantly, FACS analysis demonstrated that in vivo myeloid cell development was largely normal in PLCγ double-deficient mice compared to control mice. In vitro cytokine-driven colony formation assay confirmed that PLCγ double deficiency had no effect on SCF- or M-CSF-induced colony formation. These data were included in the Results section (pages 5, 6 and **Supplementary Fig 1f**) of the revised manuscript.

6 Similar questions arise from the results of the acute in vitro deletion of PLCs and their ensuing cell productions on OP9 and IL7. Again, the non-deficient controls, developing the expected B-lineage cells, developed some myeloid cells, although it would, again, be good to see more myeloid markers. Again, the double PLCg deficiency, blocking IL7-stimulated B-cell development, appears to have allowed (increased??) myeloid (which) development? Again, as suggested above, myeloid development-inducing culture conditions should be used to test a possible (new) preponderance of the double deficient cells for myeloid development.

FACS analysis demonstrated that myeloid cell development was largely normal in PLCγ double-deficient mice compared to control mice. In addition, in vitro cytokine-driven colony formation assay confirmed that PLCγ double deficiency BM progenitors responded normally to SCF or M-CSF. However, the possibility that PLCγ double deficiency may subtly influence development and functions of myeloid cells still exist. Further detailed studies are required to reveal any subtle effect of PLCγ double deficiency on myeloid cells. Our current study mainly focused on the role of the PLCγ pathway in IL-7-mediated early B cell development and has already yielded vast amounts of data and several novel findings. Nonetheless, the data regarding myeloid development were included (**Supplementary Fig 1** and **Fig 2b**) in the revised manuscript.

7 Although the authors show that the first defect in B-cell development occurs at the pre-proB-cell stage and involves signaling of IL7 via the IL7Ra, they find, that the small number of more differentiated early and late proB-cells (also called preBI-cells), which still develop from the already defective pre-proB-cells in vivo are also defective: they do not proliferate and do not establish preB-cell lines, as PLCg-non-defective proB- (also called preBI-cells)

do. Why? Is this an indication for the PLCg-dependent IL7 signaling at a later developmental stage of B-cells? This should be discussed.

Although IL-7 deficiency arrests B cell development largely at the pre-pro- to pro-B cell transition, CLPs, pre-pro-B and pro-B cells are all regulated by IL-7 (Nature 1998, 391:904; JEM 1994, 180:1955; 2002, 196:705). Pro-B cells depend on IL-7 for expansion and thus, IL-7- or IL-7R-deficient mice display a severe reduction of pro-B cells. PLC γ double deficiency impaired the expansion of pro-B cells, leading to the arrest of B cell development at the pre-pro-B cell stage. Nonetheless, the remaining residual PLC γ double-deficient pro-B cells were indeed defective in response to IL-7. This important point was discussed (pages 25-27) in the revised manuscript.

8 Lines 227-239: Since the pre-proB-cells of double PLCg-deficient mice are defective in development to more mature stages, it appears that the few pro(preBI)-cells, which still are formed, retain the very same defect of IL7 signaling.. The authors try to investigate, whether the decrease in numbers of more differentiated B-lineage cells could be the result of a defect of the ability to proliferate and/or to survive. IL7 has been seen to induce increased survival of pro(preBI)-cells, while the additional action of stromal cells, and kit-ligand provided by them, stimulates their proliferation. The experiments lack important controls: cells with either only IL7, or only stromal cells. It appears, therefore, premature to expect the PLCg-deficiency, and the actions of the PLCg-inhibitors to be merely, or at all, affecting proliferation. Also, myeloid differentiation results need quantification and specifications of states of differentiation.

PLC γ double-deficient BM cells failed to give rise to B cell progenitors when cultured with IL-7 alone. In contrast, PLC γ single-deficient and wild-type BM cells generated B cell progenitors when cultured in the presence of IL-7 alone. However, PLC γ double-deficient, PLC γ single-deficient and wild-type BM cells all failed to generate any B cells when cultured with OP9 stromal cells alone. Thus, IL-7 is the major driving force leading to B cell progenitor production in the OP9 in vitro culture system. These data was stated in the Results section (page 12) of the revised manuscript. In addition, myeloid differentiation was further analyzed with additional cell surface markers and cell numbers were quantified. These data were included in **Supplementary Fig 1** and described in the Results Section (pages 5 and 6) the revised manuscript.

9 Lines 240-271: The same critique holds for the experiments, where wild-type lin- BM progenitors are tested with PLCg-inhibitors and Jak3-inhibitors in these in vitro cultures. Furthermore, a more detailed marker analysis of the IL7R+ cells (lines 264 ff), which are expected to be heterogeneous and may not all react the same way, and the cells after in vitro treatments is needed to evaluate these experiments.

For inhibition experiments, Lin⁻ BM cells from wild-type or Rag1-deficient mice were cultured with OP9 cells in the presence of IL-7 for the indicated days. Then, these B cell progenitors were cultured without IL-7 or with IL-7 in the absence or presence of the indicated inhibitors. The effect of PLC γ pathway inhibition on IL-7-mediated B cell production, proliferation and survival was examined. Therefore, the inhibition experiments were performed with all necessary controls. Again, PLC γ double-deficient, PLC γ single-deficient and wild-type BM cells all failed to generate B cells when cultured with OP9 stromal cells alone as stated on page 12 of the revised manuscript.

In addition, the B cell progenitors derived from wild-type Lin⁻ BM cells following OP9/IL-7 culture without or with the indicated inhibitors were FACS analyzed. In the absence of inhibitors, these B cell progenitors were largely B220⁺CD43⁺BP-1⁻CD24⁺ early pro-B cells. The Jak3 inhibitor CD-690550 or the PLC γ inhibitor U73122 dramatically reduced the expansion of these pro-B cells. These data were added to **Supplementary Fig 6a** and described in the Results section (page 12) of the revised manuscript.

10 Lines 313-354: It is trendy, but still comes as a surprise, that the authors studied IL7-dependent metabolism. The critique of incomplete characterization of the studied IL7R+ cells, and the in vitro conditions (IL7 only, IL7+OP9, OP9 only) also applies here. Thus, the conclusions drawn from these experiments on lines 339 and 340 are at least premature. As stated above, the RAG-deficient cells studied here are CD19+pro(preB1)-cells. The data on phosphorylation of PLCg1, PLCg2, ribosomal protein S6 and mTOR are central results to the claims of the authors, that IL7 uses this pathway, here in pro(preB1)-cells. Again, OP9+IL7 and OP9 alone are missing conditions. The authors should mention, at least at this point of their studies, that the small number of PLCg-deficient pre-proB-cells remaining in the transplanted hosts for biochemical studies of the type they present, are inferior to the results with RAG-deficient pro(preB1)-cells. Again, OP9+IL7 and OP9 alone are missing conditions.

FACS analysis showed that IL-7 culture-derived Rag1-deficient IL-7R⁺ cells were largely B220⁺CD43⁺IgM⁻ pro-B cells (fraction B). Following treatment with the PLC γ inhibitors or the Jak3 inhibitor, Rag1-deficient pro-B cells displayed markedly reduced numbers and an impaired IL-7-mediated metabolism. As we addressed #8 and #9 concerns by the reviewer, we found that Rag1-deficient BM largely gave rise to B220⁺CD43⁺IgM⁻ pro-B cells when cultured with IL-7 alone or OP9 plus IL-7. Rag1-deficient BM cells failed to give rise to B cells when cultured with OP9 stromal cells alone. This information is provided in the Results Section (page 12) of the revised manuscript. Moreover, inhibition of the PLC γ pathway impaired IL-7-mediated metabolism of B cell progenitors sorted from wild-type mice without in vitro expansion (**Figs 5c,d**). Taken together, these data support the conclusion that inhibition of the PLC γ pathway or Jak3 impairs IL-7-mediated metabolism of B cell progenitors.

As we addressed #3 concern of the reviewer, PLC γ 1/2 double deficiency arrested early B cell development at the pre-pro-B cell stage whereas RAG deficiency blocks B cell development at the pro-B cell stage. Rag1-deficient pro-B cells can be readily expanded in vitro and are suitable for the study of IL-7-mediated signaling and functions. As suggested by the reviewer, we mentioned that both pre-pro- and pro-B cells respond to IL-7 stimulation, but their responses might have some differences (pages 25 and 26). This important point has been discussed (pages 25-27) in the revised manuscript.

11 Lines 384-end of results section: here, RAG-deficient pro(preB1)-cells are the target of IL7-induced responses – unfortunately again without OP9 stromal cells alone or together with IL7 - in well-conducted, convincing studies. The reviewer admits lack of expertise to judge e.g. the target specificity of these inhibitors. The reviewer is aware of studies, in which genetic mutations in the signaling pathways are used to imply a particular molecular target as operative – but it is unreasonable to suggest such mutational analyses of the PLCg1/PLCg2-DAG-mTOR.s6 pathway for the present work.

As we addressed #10 concern raised by the reviewer, Rag1-deficient BM largely gave rise to B220⁺CD43⁺IgM⁻ pro-B cells when cultured with IL-7 alone or OP9 cells plus IL-7. Rag1-

deficient BM failed to produce B cells when cultured with OP9 cells alone. Thus, the biochemical studies were performed with pro-B cells derived from Rag1-deficient BM cultured with OP9 cells plus IL-7.

We agree that the inhibitors are not absolutely specific. However, we examined the effect of three PLC inhibitors and other different inhibitors of PLC downstream effectors. These inhibitor studies showed that inhibition of the PLC γ pathway impaired IL-7-mediated functions in B cell progenitors. To overcome the specificity problem of these inhibitors, we confirmed our findings by extensive studies of primary PLC γ 1/PLC γ 2 double-deficient B cell progenitors. Especially, our RNA-seq analysis of primary mutant B cell progenitors demonstrated that PLC γ 1/PLC γ 2 double deficiency impairs proliferation, differentiation and, possibly, survival of B cell progenitors. Our studies of primary PLC γ 1/PLC γ 2 double-deficient B cell progenitors strengthen our conclusion that IL-7R controls early B lymphopoiesis through novel PLC γ -dependent activation of mTOR.

Moreover, as we addressed #1 concern of reviewer #1, a recent study demonstrates that B cell-specific deletion of Raptor blocks early B cell development at the pre-pro-B cell stage, resulting in the reduction of the number of early pro-B (fraction B) (J Immunol 2016, 197:2250). We have discussed the newly published findings in the Discussion section (page 28) of the revised manuscript.

Reviewer #3:

Reviewer #3 gives many positive comments about our manuscript. In brief, s/he thinks that our study is extensive and thorough, provides new insights into the mechanism of B cell development and IL-7R signaling, and help to understand the root of some B cell deficiencies and B cell leukemia. The reviewer also gives several specific suggestions for us to strengthen our manuscript. We have revised our manuscript based on the important suggestions.

Response to specific comments:

1) The authors use high-throughput sequencing to determine the usage frequency of VHJ558 family genes relative to VH7183 family genes (Fig. 2f). The use of these data to support the conclusion that PLC γ 1/2 double deficient pro-B cells have defective IL-7 responses is appropriate. However, readers and B cell biologists in particular may be disappointed by the scarce use of these data, and might be interested in knowing whether PLC γ 1/2-deficient pro-B cells display a general VDJ recombination defect or not. This knowledge provides a clearer overall picture of the contribution of the PLC γ pathway in early B cell development. Was the VH7183 usage in PLC γ -deficient pro-B cells normal or decreased relative to PLC γ -sufficient pro-B cells? Is this similar to that of Jak3^{-/-} pro-B cells?

We thank the reviewer for the positive comment. IL-7R-deficient pro-B cells have normal rearrangement of proximal V_H7183 segments but impaired rearrangement of distal V_HJ558 segments (Nature 1998, 391:904). Our current studies demonstrated that compared to the controls, PLC γ 1/2 double-deficient pro-B cells had normal recombination of V_H7183 family but reduced rearrangement of V_HJ558 segments. This defect was similar to that in Jak3-deficient pro-B cells. This data was included in **Fig 2f** of the revised manuscript.

2) Conclusions from experiments in which B cell progenitors were treated in cultures with PLC γ and PKC inhibitors (Figs. 3 and 5) would be greatly strengthened by using control inhibitor(s) for a pathway irrelevant in these cells. Sometimes inhibitors have unspecific effects on cell survival, proliferation and/or metabolism and showing that an inhibitor for an

unrelated pathway does not affect these processes will strengthen the conclusions of experiments that used inhibitors for the pathway of interest. There is also an issue with specificity of the inhibitors: would that be possible to show that U73122, Edelfosine and Manoadide do indeed inhibit PLC γ more directly? Do they at least inhibit Ca⁺⁺ release in these pro-B cells?

We thank the reviewer for the important suggestion. Of note, we examined the effect of the AKT inhibitor MK-2206 on IL-7-induced B cell production and mTOR activation. MK-2206 failed to block IL-7-dependent B cell production or mTOR activation. In contrast, MK-2206 markedly impaired BCR-mediated mTOR activation in splenic B cells. These data are included in the revised manuscript (**Figs. 8a,b** and **Supplementary Fig. 13**). In addition, Ca²⁺ chelator BAPTA alone barely inhibited IL-7-induced phosphorylation of S6 and S6K (**Fig. 7a**) and had little effect on IL-7-mediated B cell development in vitro (**Figs. 7c,d**). Moreover, U73122, Edelfosine or Manoadid pre-treatment had no effect on IL-7-induced phosphorylation of Stat5 in these B cell progenitors (**Fig. 6e** and **Supplementary Fig. 9a**). In contrast, CP690550 reduced IL-7-induced phosphorylation of S6, S6K and Stat5 in these B cell progenitors (**Fig. 6f**). These data demonstrate that the PLC γ inhibitors are relatively specific.

To further demonstrate the specificity of these PLC inhibitors, we examined the Ca²⁺ flux analysis in splenic mature B cells upon crosslink of BCR by anti-IgM antibodies in the presence of different concentrations of PLC inhibitors. The data showed that the PLC inhibitors impaired BCR-induced Ca²⁺ flux, demonstrating these inhibitors could directly blocked the activity of PLC γ s. These data are presented in **Supplementary Fig. 9b** and described in the Results section (page 18) of the revised manuscript.

3) The biochemical analyses are exceptionally well done, especially when considering the limited amount of pro-B cells available for these studies. However, analyses of PLC γ 1/2 double deficient pro-B cells are too limited. It is understandable that the amount of these cells is prohibitive for western blots, but it is unclear why the investigators did not take advantage of phosphoflow studies. Phosphoflow analyses of pS6 are quite robust and it would be important to show pS6 levels in PLC γ 1/2 double deficient pro-B cells treated or not with IL-7. These experiments should be feasible.

We thank the reviewer for pointing out that our biochemical analyses are exceptionally well done with the limited amount of pro-B cells and for suggesting us to detect phospho-S6 by phospho-flow cytometry analysis. We indeed tried to use phospho-flow cytometry assay to detect phospho-S6 in pro-B cells. Due to unknown reason, IL-7-induced phospho-S6 could not be detected in pro-B cells by phospho-flow cytometry. Nonetheless, we could detect IL-7-induced phospho-S6 in pro-B cells by direct Western blot analysis (**Fig 6b**). Using direct Western blot analysis, we were also able to find that the basal level of S6 phosphorylation was markedly decreased in the residual PLC γ 1/PLC γ 2 double-deficient relative to the corresponding control B cell progenitors (**Fig. 6c**). In addition, the PLC γ inhibitors reduced IL-7-induced phosphorylation of S6 in pro-B cells (**Fig. 6e**). The PKC agonist PDBu markedly induced phosphorylation of S6 (**Fig. 7b**). Taken together, these data strongly support the conclusion that IL-7 activates mTOR in B cell progenitors through the PLC γ /PKC pathway.

4) The results of experiments showed in Fig. 7b,h in which Rag1^{-/-} pro-B cells treated with the PKC agonist PDBu augment S6 and mTOR activation and cell proliferation complement greatly the inhibitor studies. Following this line of thoughts and considering PKC is

downstream of PLC γ in the pathway uncovered in this study, it should be possible to treat PLC γ 1/2 double deficient pro-B cells with PDBu (+/- ionomycin) and find increase of pS6 (by flow for instance), and cell survival and proliferation independent of IL-7.

Due to scarcity of the remaining PLC γ 1/PLC γ 2 double-deficient B cell progenitors, we were only able to isolate a small number of mutant progenitors for in vitro proliferation assay. PDBu induced cell proliferation of both wild-type and PLC γ 1/PLC γ 2 double-deficient B cell progenitors to the same extent. Thus, activation of the DAG/PKC-dependent pathway could overcome PLC γ 1/2 double deficiency to induce cell proliferation independent of IL-7. These data are presented in **Fig 7i** and described in the Results section (page 21) of the revised manuscript.

REVIEWERS' COMMENTS:

Reviewer #2 (Remarks to the Author):

My criticisms and suggestions have all been dealt with in satisfactory ways, and I now find the two messages of this one paper even more convincing, novel and highly important for a deeper molecular understanding of B cell development, and the roles of IL7 in it.

Reviewer #3 (Remarks to the Author):

The responses of the authors to all reviewers' comments are excellent and exhaustive, as they addressed every critique with additional experiments and data. A remaining issue is that the manuscript, which was already long and cumbersome to read, got even longer. I think it is up to the Editors to decide how to address best this issue.

Reviewer#1:

Reviewer #1 was unable to re-review your manuscript; however, a member of this reviewer's lab has commented in private comments to the editors that our revisions in response to reviewer 1 are satisfactory.

Reviewer#2:

Reviewer #2 thinks that her/his "criticisms and suggestions have all been dealt with in satisfactory ways." S/he "now finds the two messages of this revised paper even more convincing, novel and highly important for a deeper molecular understanding of B cell development, and the roles of IL7 in it."

Reviewer #3:

Reviewer #3 thinks that our responses to all reviewers' comments are excellent and exhaustive, as we addressed every critique with additional experiments and data. A remaining issue is that the manuscript, which was already long and cumbersome to read, got even longer. S/he thinks it is up to the Editors to decide how to address best this issue. We agree with the Editors to keep the manuscript not split.